


# Calibrated ensemble forecasts of the height of new snow using quantile regression forests and Ensemble Model Output Statistics

Guillaume Evin[1], Matthieu Lafaysse[2], Maxime Taillardat[3], and Michaël Zamo[3]

[1]Univ. Grenoble Alpes, INRAE, UR ETGR, Grenoble, France
[2]Univ. Grenoble Alpes, Université de Toulouse, Météo-France, CNRS, CNRM, Centre d'Etudes de la Neige, 38000 Grenoble, France
[3]CNRM, Université de Toulouse, Météo-France, CNRS, Toulouse, France

**Correspondence:** Guillaume Evin (guillaume.evin@inrae.fr)

**Abstract.** Height of new snow (HN) forecasts help to prevent critical failures of infrastructures in mountain areas, e.g. transport networks, ski resorts. The French national meteorological service, Meteo-France, operates a probabilistic forecasting system based on ensemble meteorological forecasts and a detailed snowpack model to provide ensembles of HN forecasts. These forecasts are however significantly biased and underdispersed. As for many weather variables, post-processing methods can

be used to alleviate these drawbacks and obtain meaningful 1-day to 4-day HN forecasts. In this paper, we compare the skill of two post-processing methods. The first approach is an ensemble model output statistics (EMOS) method, which can be described as a Nonhomogeneous Regression with a Censored Shifted Gamma distribution. The second approach is based on quantile regression forests, using different meteorological and snow predictors. Both approaches are evaluated using a 22-year reforecast. Thanks to a larger number of predictors, the quantile regression forest is shown to be a powerful alternative

to EMOS for the post-processing of HN ensemble forecasts. The gain of performance is important in all situations but is particularly marked when raw forecasts completely miss the snow event. This type of situations happens when the rain-snow transition elevation is overestimated by the raw forecasts (rain instead of snow in the raw forecasts) or when there is no precipitation in the forecast. In that case, quantile regression forests improve the predictions using the other weather predictors (wind, temperature, specific humidity).

## 1   Introduction

In cold regions (e.g. mountainous areas), the height of new snow (HN) expected for short lead times is critical for many safety issues (e.g. avalanche hazard) and the economical impacts of dysfunctional transport networks (road, airports, train tracks viability). National weather services increasingly provide automatic predictions in that purpose, usually relying on numerical weather prediction (NWP) model outputs. Forecasting HN is particularly challenging for many reasons. First, the precipitation

forecasts in NWP models are significantly biased and underdispersed. Then, HN is strongly dependent on elevation in mountainous areas and this relationship can not be perfectly reproduced by the current resolution of NWP models. Finally, several processes affecting snow properties (density, height, precipitation phase) are either absent or poorly represented in NWP mod-





els (e.g. density of falling snow, mechanical compaction during the deposition). In particular, the evolution of the rain–snow limit elevation can greatly differ according to meteorological conditions and is only partly understood (Schneebeli et al., 2013).

Few attempts have been made to post-process ensemble HN forecasts. To the best of our knowledge, Stauffer et al. (2018) and Scheuerer and Hamill (2019) are the first studies to present post-processed ensemble forecasts of HN. They consider direct ensemble NWP outputs as predictors (precipitation and temperature). Nousu et al. (2019) incorporate physical modelling of the snowpack in order to integrate the high temporal variations of temperature and precipitation intensity during a storm event, which can have highly non-linear impacts on HN. In addition, Nousu et al. (2019) demonstrate the ability of a nonhomogeneous

regression method to improve the ensemble forecasts of HN from the PEARP-S2M ensemble snowpack. Using a regression method based on the censored shifted Gamma distribution (Scheuerer and Hamill, 2015, 2018), the forecast skill was improved for the majority of the stations from common events to more unusual events. However, as this method only considers a single predictor (the simulated HN itself), on a given point, dry days and rainy days can not be discriminated as long as all forecast members provide a zero value for HN. This prevents from an appropriate correction of some specific NWP errors, such as a

systematic error among all simulation members in the rain-snow transition elevation.

    In this study, we consider the application of quantile regression forests (QRF) as an alternative to nonhomogeneous regression methods. This approach has been successfully applied for the post-processing of ensemble forecasts of surface temperature, wind speed (Taillardat et al., 2016) and rainfall (Taillardat et al., 2019). QRF are often considered as a non-parametric method since it does not rely on an explicit mathematical relationship between the predictors and the target distribution of the

predictand. Furthermore, many predictors can be incorporated without decreasing the forecast skill (Taillardat et al., 2019), which can be particularly interesting in our case when, for example, the raw ensemble only contains zero HN while rainfall forecasts are large. Indeed, in some cases, the PEARP-S2M ensemble snowpack completely misses important snow events (e.g. due to an erroneous rain/snow limit). For some problematic meteorological situations, QRF can possibly provide a specific correction.

Section 2 summarizes the forecasts and observations dataset used in this study. Section 3 provides the details of the ensemble model output statistics (EMOS) method tested in this study, a particular nonhomogeneous regression method already employed in Nousu et al. (2019) and considered here as a benchmark method. Section 4 describes the QRF method. In Section 5, we detail the evaluation of the performances of each method. Section 6 presents the results. Finally, Section 7 provides a discussion of the results with some outlooks.

## 2   Data

In this study, we select 92 stations in the French Alps and Pyrénées based on a minimum availability of observations of 60% (percentage of missing observations thus varies between 0% and 40% with an average of 18%). Forecasts and observations are available and reliable for these 92 stations presented in the Figure 1, for 22 winter seasons covering the period 1994-2016, where each winter season starts on the 6th of December and ends on the 30th of April of the following year (3218 days in

total).



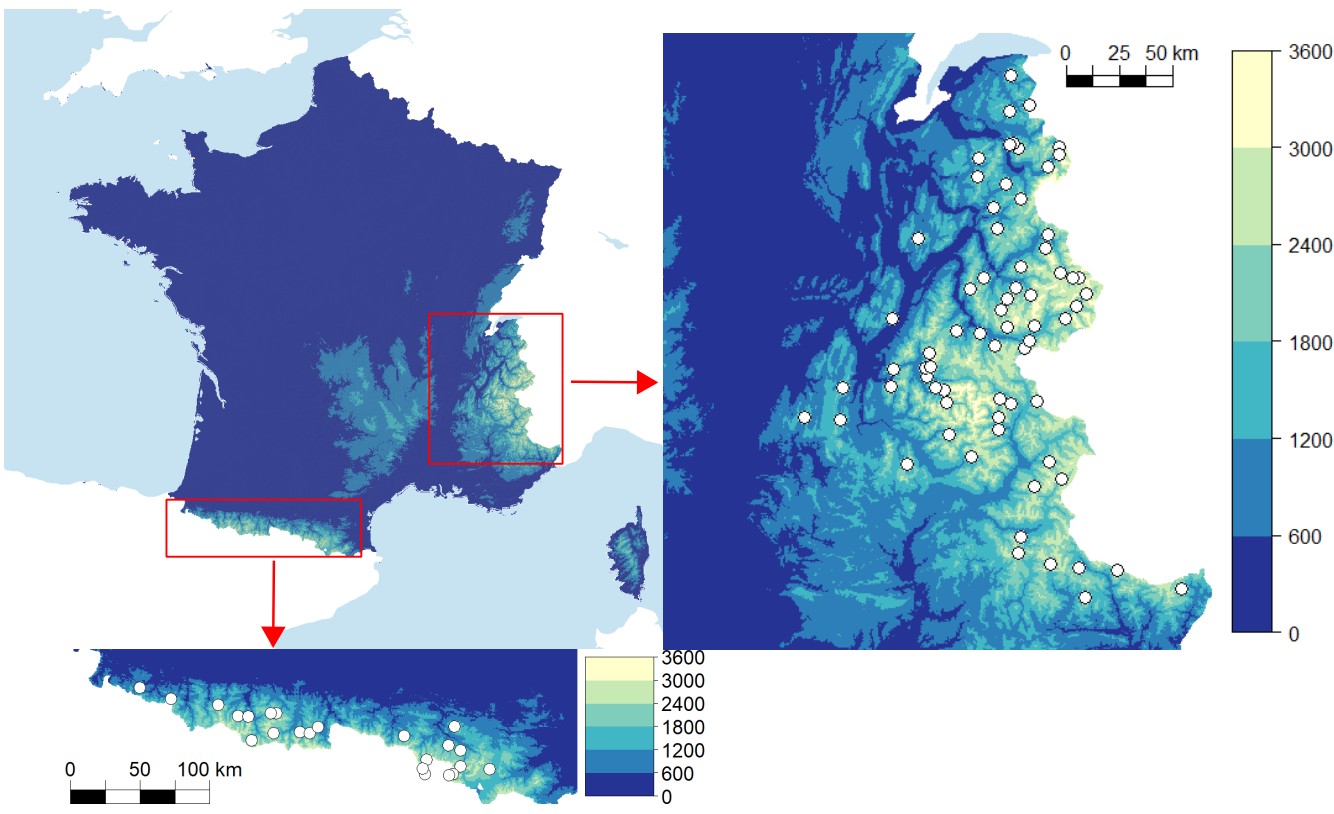

**Figure 1.** Map of the 92 observation stations (white dots) in the French Alps (right plot) and Pyrenees (bottom plot).

The forecasts are obtained by a chain of ensemble numerical simulations. 10-member reforecasts of the PEARP ensemble NWP (Descamps et al., 2015; Boisserie et al., 2016) are downscaled by the SAFRAN system (Durand et al., 1999) to obtain a meteorological forcing adjusted in elevation. The Crocus multilayer snowpack model, part of the SAFRAN - SURFEX/ISBA-Crocus - MEPRA (S2M) modelling chain (?Vernay et al., 2019), is forced by these forecasts to provide ensemble simulations of HN accounting for all the main physical processes explaining the variability of HN for a given precipitation amount: the dependence of falling snow density on meteorological conditions, the mechanical compaction over time depending on snow weight, the microstructure and wetness of the snow, a possible surface melting, and so on. The forecasts used in this paper are the same as used by Nousu et al. (2019) who provided more details on the models' configurations.

Table 1 presents the selected predictors based on the available reforecasts. This selection is derived from studies of Scheuerer and Hamill (2015) and Taillardat et al. (2019) for rainfall. It includes summary statistics and probabilities of the variable to be predicted (rainfall in the previous references transposed into HN in our case). We also consider statistics on other weather variables of the ensemble suspected to add predictability because they might affect the statistical relationship between observed and simulated HN.


**Table 1.** Set of all available predictors.

| Name | Description |
|------|-------------|
| CTRL | Control member of raw ensemble of HN |
| MEAN | Mean of raw ensemble of HN |
| MED | Median of raw ensemble of HN |
| Q10 | First decile of raw ensemble of HN |
| Q90 | Ninth decile of raw ensemble of HN |
| PR0 | Raw probability of HN |
| PR1 | Raw probability of HN > 1 cm |
| PR3 | Raw probability of HN > 3 cm |
| PR5 | Raw probability of HN > 5 cm |
| PR10 | Raw probability of HN > 10 cm |
| PR20 | Raw probability of HN > 20 cm |
| SIGMA | Standard deviation of raw ensemble of HN |
| IQR | IQR of raw ensemble of HN |
| q10, 50, 90 are the first decile, the median, and ninth decile of | |
| the raw ensemble for these variables: | |
| SNOWR_q10,50,90 | Snow rate [kg m$^{-2}$ h$^{-1}$] |
| RAINR_q10,50,90 | Rain rate [kg m$^{-2}$ h$^{-1}$] |
| WIND_q10,50,90 | Wind speed [m/s] |
| TAIR_q10,50,90 | Near Surface Air Temperature [K] |
| QAIR_q10,50,90 | Near Surface Specific Humidity [ø] |

In this paper, for each station, we thus consider $i = 1, \ldots, n = 3218$ days with an observed HN $Y_i$ (the response) and a vector

of corresponding predictors $\mathbf{X}_i$.

## 3   Ensemble model output statistics

Among the Ensemble model output statistics (EMOS) methods available, non-homogeneous regression approaches are the most common and were originally based on Gaussian regressions whose mean and variance are linear functions of ensemble statistics (Gneiting et al., 2005; Wilks and Hamill, 2007). Non-homogeneous Regression methods can also incorporate climatological

properties, and additional predictors. For meteorological predictands such as rainfall and snow, however, the high number of zero-values motivate the use of a discrete-continuous distribution with a mass of probability in zero. In this study, we use a regression based on the Zero-censored Censored Shifted-Gamma distribution (CSGD, see Scheuerer and Hamill, 2015, 2018; Nousu et al., 2019).



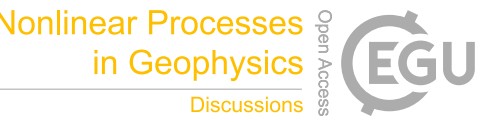

The non-homogeneous regression method applied in this study is similar to the approach presented in Nousu et al. (2019) and is referred to as EMOS hereafter. Further details of this EMOS method are presented in Appendix A. More precisely, we detail how the CSGD is used to represent the predictive distribution of daily HN forecasts, the parameter estimation method, and the related predictive distribution. Please note that expression (A3) is slightly different from expression (4) in Nousu et al. (2019) and strictly corresponds to Scheuerer and Hamill (2018) (see their expression of $\sigma$ in section 3a, p. 1653). While this difference is not critical on the performances, expression (A3) avoids scaling issues in parameter $\beta_2$.

## 4 Quantile regression forest

Compared to the EMOS method, quantile regression forest is expected to incorporate any predictor without degrading the quality of the predictions. Subsets of the space covered by the predictors are created in order to obtain homogeneous groups of observations inside these subsets. If the predictors include many meteorological forecasts, these subsets are expected to describe different meteorological situations. Compared to EMOS, this so-called non-parametric regression does not assume a particular distribution for the predictors or the response, and empirical distributions represent the uncertainty about the prediction.

### 4.1 Method

The QRF method presented in this paper are based on the construction of binary decision trees, as proposed by Meinshausen (2006). These decision trees (CART; Breiman et al., 1984) are built by iteratively splitting each predictor space ($\mathscr{D}_0$) into two groups ($\mathscr{D}_1$ and $\mathscr{D}_2$) according to some threshold. The predictor and the threshold or grouping are chosen in order to maximize the homogeneity of the corresponding values of the response (here the observed HN) in each of the resulting groups. Homogeneity is defined as the sum of variances of the response variable within each group:

$$\nu(\mathscr{D}_j) = \sum_{Y \in \mathscr{D}_j} \left[ Y - \bar{Y} \right]^2,$$

where $Y$ and $\bar{Y}$ corresponds to the response sample and its mean in $\mathscr{D}_j$, respectively. The optimal threshold $s$ maximizes:

$$\mathscr{H}(\mathscr{D}_1, \mathscr{D}_2) = \max_{s \in \mathscr{T}^*} \left[ \nu(\mathscr{D}_0) - \nu(\mathscr{D}_1) - \nu(\mathscr{D}_2) \right],$$

where $\mathscr{T}^*$ is a random subset of the predictors in the predictors' space $\mathscr{T}$. These trees are obtained by bootstrapping the training data, which justifies the name of "random forest" since each split of each tree is built on a random subset of the predictors (Breiman, 2001). The final "leaf" corresponds to the group of predictors at the end of each tree (see Fig. 1 in Taillardat et al., 2019, for an illustration).





### 4.2 Implementation

105 The QRFs are obtained using the function `quantregForest` of the package `quantregForest` in R (R, 2017). The splitting procedure described above can be constrained by different choices, e.g. a minimum number of observations in leaves. In this paper, we grow 1000 trees which represent a sufficiently large number of trees to span a great variety of meteorological situations without demanding an unbearable computational effort. Different values for the parameters `mtry` and `nodesize` have been tried, the best performances being found for `mtry=2` and `nodesize=10` (see Section 6).

### 110 4.3 Predictive distribution

For QRF, the predictive distribution given a new set of predictors $x$ is the conditional CDF introduced by Meinshausen (2006):

$$\hat{F}(y|x) = \sum_{i=1}^{n} w_i(x)\mathbb{1}(\{Y_i \leq y\}), \tag{1}$$

where the weights $w_i(x)$ are deduced from the presence of $Y_i$ in a final leaf of each tree when one follows the path determined by $x$. Different quantiles can then be obtained from these predictive distributions.

### 115 5 Evaluation

This Section details the process applied to assess the performance of the different approaches. Classical evaluation metrics include the continuous ranked probability score (CRPS) which sums up the forecast performance attributes in terms of both reliability and resolution (Murphy and Winkler, 1987; Hersbach, 2000; Candille and Talagrand, 2005). Rank histograms are also a common tool to assess systematic biases and over/under dispersion.

### 120 5.1 Cross-validation

For all the experiments of this study, we use a leave-one season out cross-validation scheme. For each of the 22 seasons, the season is used as a validation dataset while the other 21 seasons are used for training. It first ensures that a robust calibration of the post-processing methods is obtained. It also avoids the evaluation of the performances with a unique validation period that could be atypical (e.g. a very snowy/dry winter season).

### 125 5.2 CRPS

The CRPS is one of the most common probabilistic tools to evaluate the ensemble skill in terms of reliability (unbiased probabilities) and resolution (ability to separate the probability classes). For a given forecast, the CRPS corresponds to the integrated quadratic distance between the cumulative distribution function (CDF) of the ensemble forecast and the CDF of the observation. Commonly, the CRPS is averaged over $n$ days:



$$\overline{\mathrm{CRPS}} = \frac{1}{n}\sum_{i=1}^{n}\int_{\mathrm{I\!R}}\big[F_i(y) - H(y - Y_i)\big]^2\mathrm{d}y, \tag{2}$$

where $F_i(x)$ is the CDF obtained from the ensemble forecasts for day $i$, $Y_i$ is the corresponding observation, and $H(z)$ is the Heaviside function ($H(z) = 0$ if $z < 0$; $H(z) = 1$ if $z \geq 0$). The CRPS value has the same unit as the evaluated variable and is equals to 0 for a perfect system.

For the EMOS-CSGD model described above, an analytic formulation of the CRPS is available (Scheuerer and Hamill, 2015) and a correct CRPS estimation is directly obtained.

In other cases, a correct evaluation of the CRPS (2) can be technical. For example, the raw ensemble forecasts do not provide the true forecast CDF, but only an ensemble of values. In this case, the CRPS is estimated with some error. In this study, we apply the recommendations given by Zamo and Naveau (2018). More specifically, when the forecast CDF is known only through an $M$-ensemble $x_1,\ldots,x_M$, we apply the following definition to estimate the instantaneous CRPS (i.e. for one ensemble):

$$\widehat{\mathrm{CRPS}}_{\mathrm{INT}} = \int_{\mathrm{I\!R}}\big[\frac{1}{M}\sum_{m=1}^{M}H(x - x_m) - H(x - y)\big]^2\mathrm{d}x, \tag{3}$$

where $y$ is the observation corresponding to the forecast ensemble. This expression is evaluated with the function `crpsDecomposition` of the R package `verification`. In the case of the instantaneous CRPS of the raw ensemble forecasts, Eq. 3 is applied directly, while some refinements can be made to improve the estimated CRPS in the case of QRF. To evaluate instantaneous CRPS values for QRF, we use the recommendations by Zamo and Naveau (2018), i.e. we use the average $\widehat{\mathrm{CRPS}}_{\mathrm{INT}}$ given in Eq. (3) with linearly interpolated regular quantiles between unique quantiles. The so-called regular ensemble of $M = 200$ quantiles $x_{i=1,\ldots,M}$ of orders $\tau_{i=1,\ldots,M} \in [0;1]$, is defined as $z_i = F^{-1}(\tau_i)$, for all $i$, with $\tau_i \in \{\frac{1}{M}, \frac{2}{M}, \ldots, \frac{M-1}{M}, \frac{M-0.1}{M}\}$.

### 5.3 Rank histograms

The reliability of ensemble forecast systems can be assessed using rank histograms (Hamill, 2001). If the predictive distributions obtained with the different post-processing methods are adequate, then the CDF values of the predictive distributions for the observations should be uniformly distributed (so-called Probability Integral Transform, PIT). The flatness of the histogram of these CDF values is a necessary but not sufficient condition of the system reliability. Systematic biases are detected by strongly asymmetric rank histograms. It is also an indicator of the spread skill as underdispersion will result in a U-shaped rank histogram and overdispersion in a bell-shaped rank histogram. Rank histograms can be computed for the whole forecast dataset, or stratified according to different classes of average ensemble forecasts (stratifying according to the observations leading erroneous conclusions, see Bellier et al., 2017). In this study, as proposed in the recent study of Bröcker and Bouallègue (2020), a stratification based on the average of the combination of raw forecasts and the verification observations is used. Three





HN intervals are considered: [0 cm ; 10 cm[, [10 cm ; 30 cm[ and [30 cm ; ∞[. To guarantee a sufficient sample size for rank histograms, they are computed for the whole evaluation dataset by considering all dates and stations as independent.

**6  Results**

We first discuss the application of the QRF methods with regards to the parameters `mtry`, which specifies the number of predictors randomly sampled as candidates at each split (usually small, i.e. less than 10) and `nodesize` which defines the minimum number of cases (days) in terminal nodes. Different values have been tried for both parameters (2, 4, 6, 8 and 10 for `mtry`, and 5, 10, 15, 20 for `nodesize`). For a 1-day lead time, the best (smallest) average CRPS values are obtained

for small values of `mtry` (2 or 4) and high `nodesize` values (15 or 20), the mean CPRS being minimized for `mtry=2` and `nodesize=10` (results not shown). However, the range of mean CRPS values is narrow (1.282 and 1.294). We conclude here that the performances obtained with the QRF approach are not very sensitive to the value of the QRF parameters and `mtry=2` and `nodesize=10` are retained in the rest of this study.

Figure 2 highlights the most important predictors for the QRF method, for different lead times. The importance criteria

here is the mean decrease in node impurity from splitting on the variable, averaged over all trees, as implemented by the function `importance` of the package `randomForest` in R. The notion of node impurity is related to an impurity measure defined here as the standard deviation of the response variable, i.e. a node purity increases when the homogeneity of the response $Y$ increases in the corresponding trees (see, e.g. Louppe et al., 2013, for further details). For a 1-day lead time, the Q90 of the forecast snow rate, followed by the Q90 of raw forecasts of HN, are the most important predictors. The most

important predictors are directly related to snow quantities, and the role of other meteorological forcings is minor. As the lead time increases, the importance of the snow predictors decreases while the importance of the forecast rain rate gets larger. In particular, the Q90 of the snow and rain rates are the two most important predictors at a 4-day lead time.

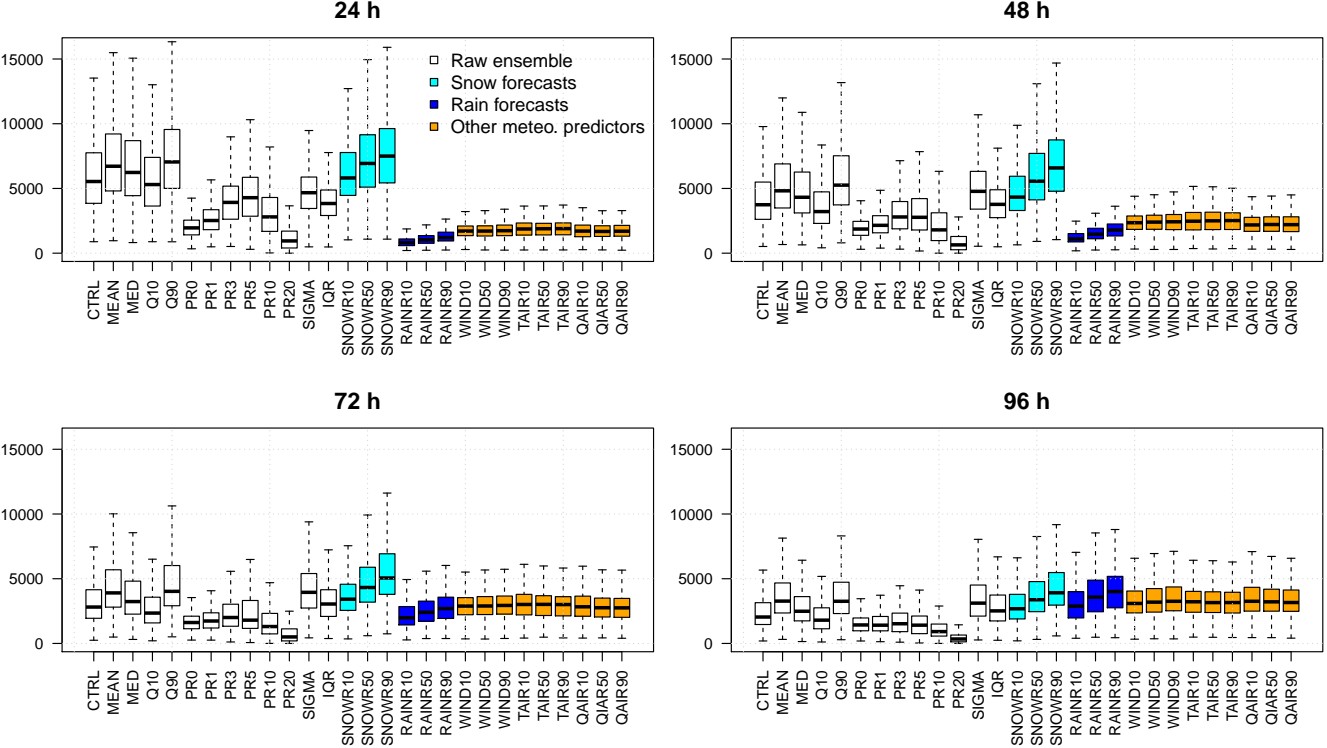

**Figure 2.** Importance of the predictors for different lead times for the QRF method.





Figure 3 shows the time series of observed HN for a period with significant snowfalls, along with raw reforecasts and predictive intervals with a 80% probability obtained with the different post-processing methods, for the station 5079400 at Le

Monêtier-les-Bains during the period 26 Feb. 2001 - 18 Mar. 2001 and for a 1-day lead time. The following observations can be made:

– The raw ensembles generally underestimate the largest observed HN (see, e.g., the period 26 Feb. - 05 Mar.). The intervals given by the raw ensembles are thin and under-dispersed in comparison to post-processed ensembles.

– Predictive intervals obtained with the post-processing methods are large and look very similar. Observations generally

lie within these intervals (with one major exception at the end of March).

– When the raw reforecasts are all equal to zero, the EMOS method mechanically predicts zero HNs, which is often verified (see, e.g. on the 5th, 6th and 11th of March). However, EMOS predicts these zero values with a 100% probability, while QRF predicts small intervals in this example, which avoids failures (i.e. prediction of a zero value with absolute certainty while a positive HN value is observed). In this example, it happens for two days, on the 7th and 9th of March.

Figure 4 shows the time series for the station 5079400 at Saint-Paul-sur-Ubaye during the period 29 Mar. 2012 - 18 Apr. 2012. An important observed HN of 40 cm occurred on the 10th of April. EMOS method completely misses this event, because no snow was present in the raw forecasts. In this case, QRF predicts a large interval, with a 90th percentile around 20 cm. Looking at the raw forecasts of the meteorological forcings for this day, the 90% CI of the snow rate is [0.7, 8.1] cm/h, [1.2, 9.4] cm/h for the rain rate and [2.1, 3.2] °C for the air temperature. High snow/rain rates combined to above-zero temperatures led to

zero HN forecasts by the snowpack model, while the QRF method exploits these high precipitation rates in order to predict significant HN amounts.



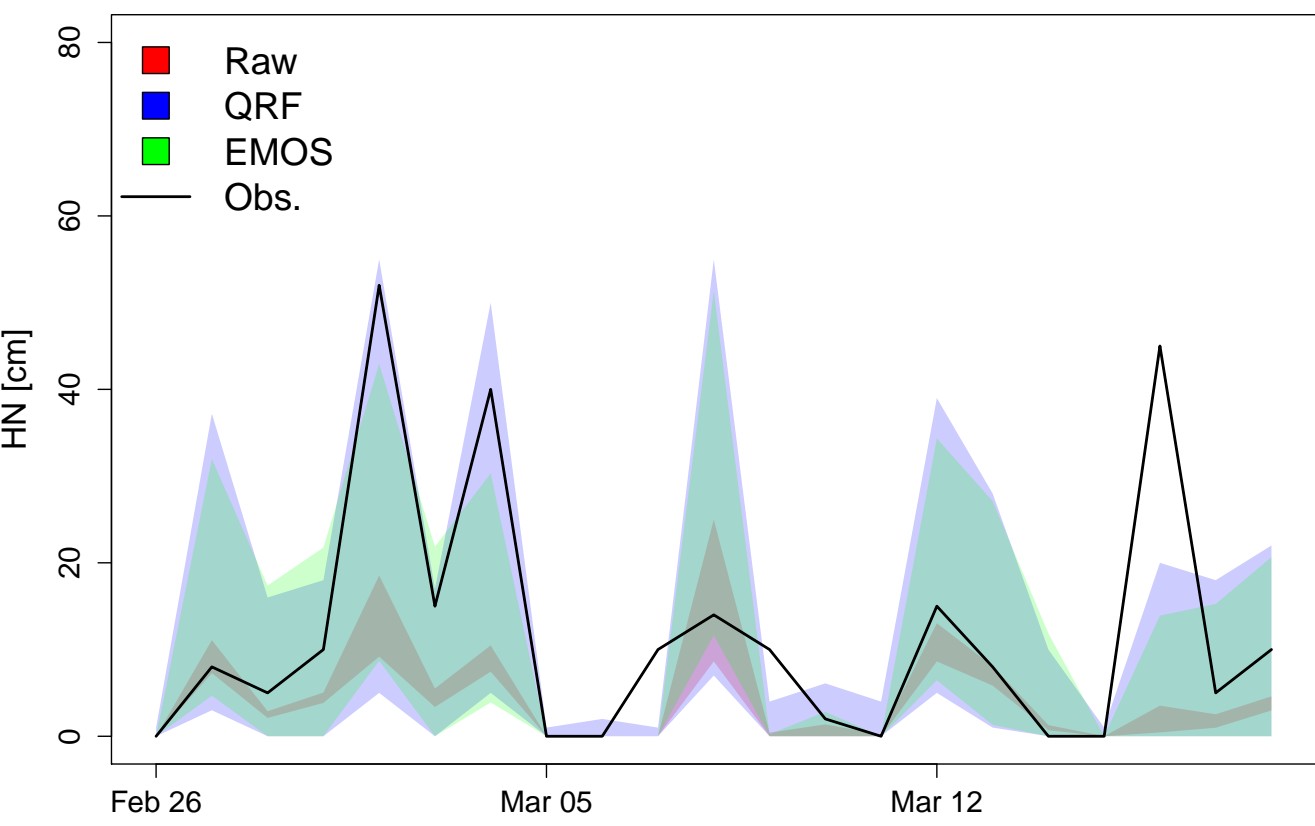

**Figure 3.** Time series of the raw reforecasts for a 1-day lead time (red) and predictive quantiles using QRF (blue) and EMOS (green) during March 2001 for the station 5079400. The envelopes represent the interval between the 10th and 90th percentiles and the solid black line represents the time series of the HN observations.



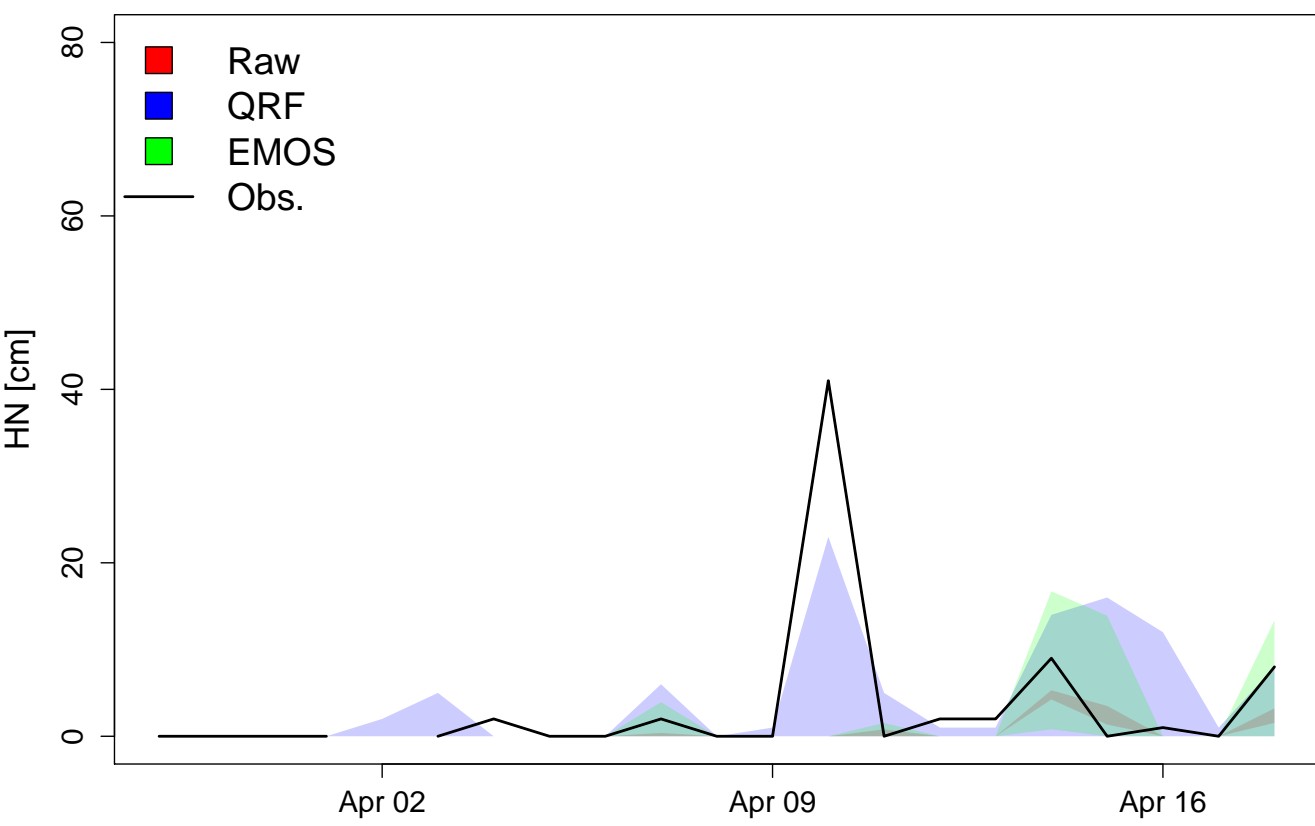

**Figure 4.** Time series of the raw reforecasts for a 1-day lead time (red) and predictive quantiles using QRF (blue) and EMOS (green) during April 2012 for the station 4193400. The envelopes represent the interval between the 10th and 90th percentiles and the solid black line represents the time series of the HN observations.


Figure 5 shows the 2024 CRPS values averaged over the different winter seasons (92 stations × 22 winter seasons) obtained with the raw reforecasts, and with EMOS and QRF post-processing methods, for a 1-day lead time (left plots). While EMOS gives a significant gain of performance, it is still outperformed by the QRF method. The second column quantifies this im-

provement as a percentage, in terms of relative CRPS. For most of the stations, QRF leads to an improvement between 20% and 30%, up to 40% compared to EMOS. Results (not shown) are very similar for the other lead times.

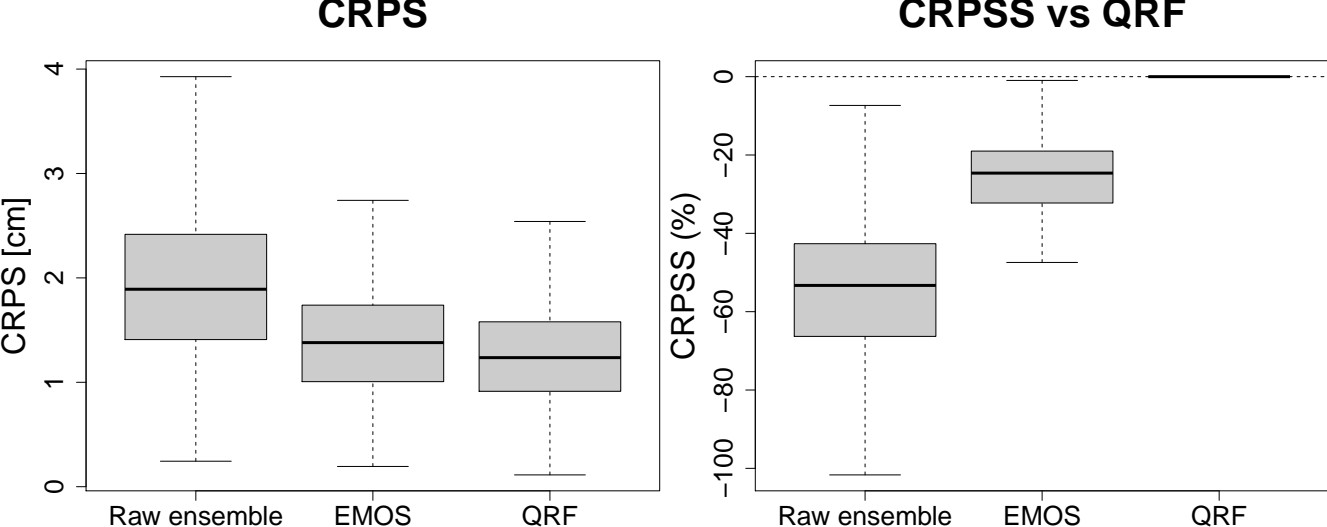

**Figure 5.** Boxplots of CRPS (left plots) and relative CPRS with QRF as a reference (right plots) with the different methods for all locations, for a 1-day lead time.





Figure 6 shows the rank histograms of HN with the raw forecasts and with EMOS and QRF post-processing methods. As indicated in previous studies (see, e.g. Nousu et al., 2019), raw forecasts are clearly underdispersed, leading to a U-shape rank histograms, and also usually underestimate large HN values (over-representation of the last class). These defaults are

particularly visible for classes of raw ensemble averages above 10 cm (rows 2-3). The rank histogram with the EMOS method is almost perfectly flat for the small ensemble/observation averages ([0, 10) cm). For larger classes of events, it seems that EMOS predictive distribution is slightly underdispersed. The QRF method shows better performances than EMOS in that regard, the only limitation being an underestimation of the largest snowfalls (see last class for HN > 30 cm in the bottom-right plot).



**Figure 6.** Rank histograms of HN forecasts for three classes of HN ensemble/observation mean, with the different methods, for a 1-day lead time.





Figure 7 shows the ROC curves (Kharin and Zwiers, 2003) which illustrate the quality of probability forecasts by relating the hit rate (probability of detecting an event which actually occurs) to the corresponding false-alarm rate (probability of detecting an event which does not occur). The ROC curves are shown for three categories: All snow events (HN greater than 1 cm, 19% of the observed cases), "moderate" snow events (HN greater than 10 cm, 5% of the observed cases) and "rare" snow events (HN greater than 30 cm, 1.4% of the observed cases).

Figure 7a shows that the raw forecast ensemble performs almost as well as post-processed ensembles when all snow events are considered. For this category, the red curve corresponding to the QRF approach deviates farther away from the no-skill diagonal than the green curve corresponding to the EMOS method, indicating a better skill of the QRF approach. For moderate snow events (Fig. 7b), while QRF and EMOS show similar performances, the ROC curve corresponding to raw ensembles is close to the diagonal and indicates almost no skill. For rare and intense snow events exceeding 30 cm of fresh snow in one day,

Fig. 7c shows a slight gain of performance with the QRF approach compared to EMOS.

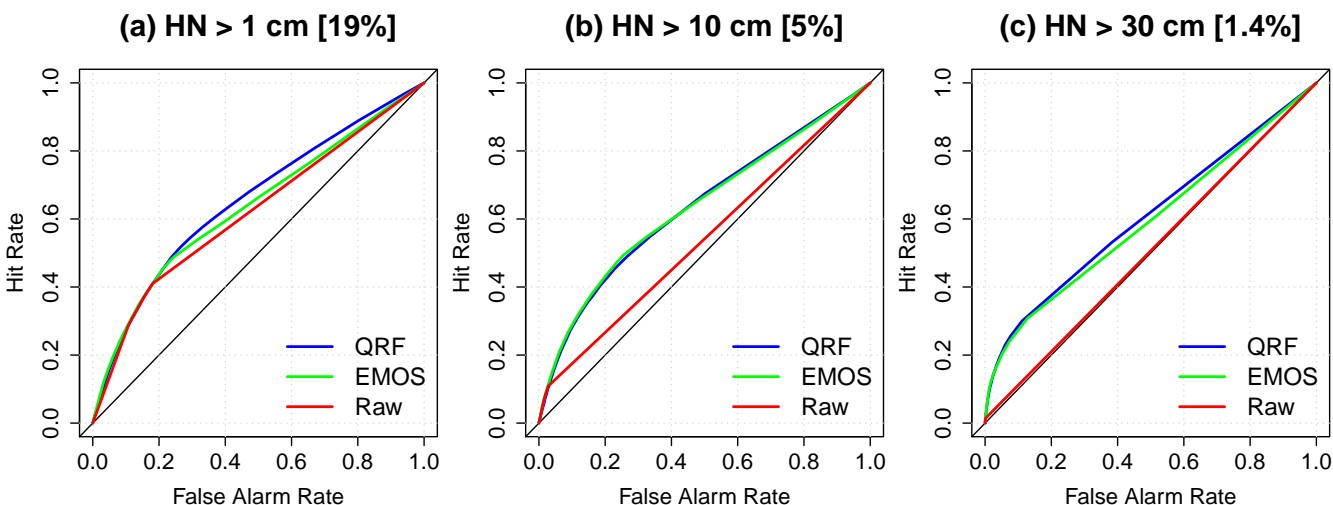

**Figure 7.** ROC curves for different snow events. (a) HN exceeds 1 cm. (b) HN exceeds 10 cm. (c) HN exceeds 30 cm. Values between brackets indicate the observed frequencies, in percent. A "good" prediction must maximize hit rate and minimize false alarms.

To investigate further the different behaviours of EMOS and QRF, Figure 8 shows the relative CRPS value of EMOS versus QRF for all dates and stations with a positive observed HN values (greater than 1 cm) and for different classes based on the predictors. More specifically, we try to investigate the difference of performances according to the presence or not of at least one positive rain/snow rate value among the different members of the ensemble forecast. Cases where there is not any rain or

snow in the forecasts while positive HN values have been observed represent only 0.4% of all dates and stations (Figure 8a). Cases corresponding to precipitation phase errors (at least one member with rain in the forecasts but no snow while a positive HN has been measured, Figure 8b) represents 1.2% of all cases. Obviously, cases with snow in the forecasts and a positive HN are more frequent (8.9% and 14.7% for cases (c) and (d), respectively). Overall, while QRF outperforms EMOS in all cases





(as outlined in Fig. 2), we see that the gain of performances is particularly marked for cases (a) and (b), i.e. when there is no
snow in the forecasts. These results demonstrate the advantage of the QRF approach in this case, i.e. when other predictors
(rain, temperature, etc.) can be exploited to overcome the limitations of the snow forecasts for the prediction of observed HN
(see further discussion in Section 7 below).

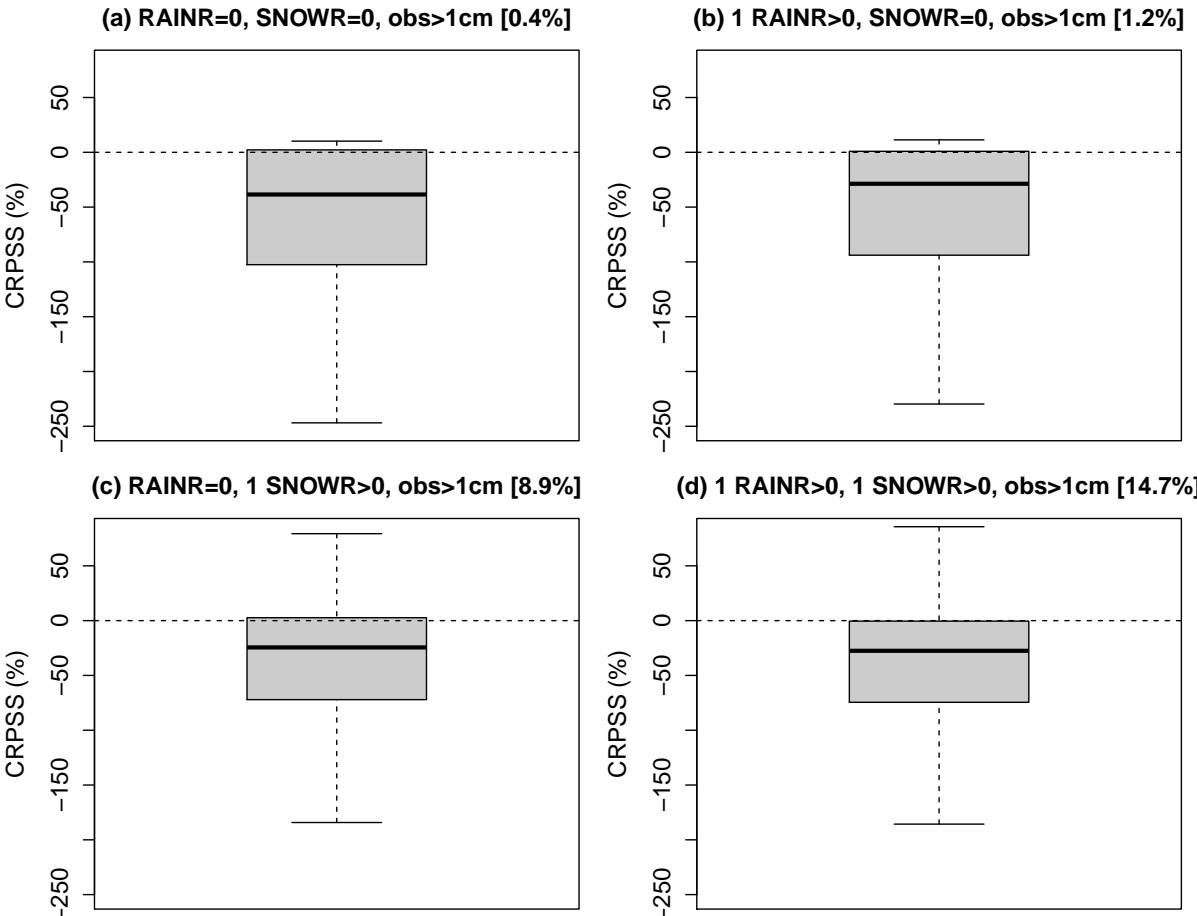

**Figure 8.** Relative CPRS values of EMOS versus QRF as percentages, with QRF as a reference, for all dates and stations and for a 1-day lead
time. Only dates with a positive observation greater than 1 cm are selected, and for different classes of predictors. (a) All forecast members
of rain rate and snow rate equals to 0. (b) All forecast members of rain rate equal to 0 and at least one member with a positive snow rate
value. (c) At least one member with a positive rain rate value and all forecast members of snow rate equals to 0. (d) At least one member with
a positive rain rate value and one member with a positive snow rate values. Values between brackets indicate the corresponding frequencies,
in percent, among all the dates.





## 7 Discussion and outlooks

### 7.1 Comparison of performances between QRF and EMOS approaches

In this paper, we compare the scores of post-processed forecasts of the 24h height of new snow between two commonly used statistical methods: EMOS and QRF. With this dataset, the added value of QRF is unambiguous with a general improvement of CRPS, an improvement of rank diagrams for severe snowfall events, and a slight improvement of ROC curves for more common events. The predictors selected by the QRF training clearly suggest that the simulated HN from the Crocus snow cover model is useful but not sufficient to optimize the post-processed forecasts as the meteorological variables forcing the

snow cover model are also selected by the algorithm. The added value coming from these meteorological predictors is the most likely explanation of the improvement obtained between QRF and EMOS. This improvement is frequent in various situations and the physical reason for which the simulated HN does not translate all the predictive power of the meteorological forcings is probably not unique, but can be partly explained by the presence of precipitation phase errors.

### 7.2 Role of precipitation phase errors in the added value of QRF

The examples selected for illustration suggest that phase errors (or in other words errors in the rain-snow transition elevation) is one of the possible explanations for the insufficient predictive power of the simulated HN. Indeed a number of observed snowfall events are simulated with a zero value in terms of HN, sometimes for all members, but with a significant precipitation amount. EMOS is not able to consider these days with a significant probability of positive HN because they are identical to dry days when considering only this predictor, whereas the other predictors considered by QRF (total precipitation, air temperature)

can help to discriminate the days with an error in phase but with forecast precipitation and relatively cold conditions from dry days or warm days. This assumption is difficult to be statistically generalized due to the large variety of situations, i.e. errors in precipitation phase often concern only a part of the total duration of a snowfall event and/or a part of the simulation members. Nevertheless, our classification of CRPSS depending on rainfall and snowfall occurrence shows a systematic improvement of CRPS by QRF for the cases where an error in the rain-snow transition elevation is the most obvious (e.g. observed snowfall

with simulated rainfall but no simulated snowfall during the whole day for all members, Fig. 8b).

The sensitivity of snow cover models to errors in precipitation phase was already illustrated by Jennings and Molotch (2019) with a meteorological forcing built from weather stations. The magnitude of errors is expected to be much higher when the forcing comes from NWP forecasts. The reduction of phase errors in atmospheric modelling is beyond the scope of this paper. However, an improvement of post-processed forecasts might be opened by considering predictors more directly related to this

phase issue. In particular, interviews of operational weather forecasters show that expert HN forecasts strongly rely on the $1°C$ isothermal level in terms of pseudo-adiabiatic wet-bulb potential temperature ($\theta'_w$). Unfortunately, this diagnostic was not available in the PEARP reforecast, but this feedback encourages future reforecast productions to include this additional diagnostic as the post-processing might be able to more directly account for phase errors with such a predictor. More simply considering the surface wet-bulb temperature is also increasingly done in land surface modelling for phase discrimination

(Wang et al., 2019) and it may also be an easier alternative predictor for statistical processing, although the information content





of the simulated atmospheric column is probably better summarized by the iso-$\theta'_w$. Nevertheless, forecasters also mention that a common limitation of NWP models is their inability to simulate the unusually thick $0°C$ isothermal layers encountered in some intense storms, up to 1000 m. The complex interactions between the processes involved in this phenomenon are only partly understood (latent cooling from melting precipitation and evaporation/sublimation, melting distance of snowflakes, adiabatic

cooling of rising air, specific topographies, blocked cold air pockets, etc. Minder et al., 2011; Minder and Kingsmill, 2013). In these specific cases, even the level $\theta'_w = 1°C$ is considered to be a poor predictor of the rain-snow transition elevation. These situations are often the most critical in terms of impacts (wet snow at low elevations affecting the roads and the electrical network) but their very low frequency will remain a severe challenge even with a statistical post-processing.

### 7.3   Limitations for operational perspectives

In order to investigate the potentials of the statistical methods themselves regardless of the constraints on the available dataset, we choose in this paper to calibrate and evaluate the post-processing methods on the same 22-year long dataset with a cross-validation scheme. However, Nousu et al. (2019) illustrate the strong impact of the discrepancies between reforecasts and operational forecasts in the post-processing efficiency. In complementary investigations (not shown), we noted that QRF is even more sensitive to the homogeneity between calibration and application datasets. For instance, the added value of QRF

compared to EMOS was completely lost when using the evaluation dataset of Nousu et al. (2019) (operational PEARP-S2M forecasts). Therefore, despite the significant added value of QRF compared to EMOS with consistent and homogeneous datasets for calibration and evaluation, its practical implementation in real-time operational forecasting products is still a challenge because reforecasts strictly identical to operational configurations are often not available. Time adaptative training based on operational systems is an alternative to favour the homogeneity of the dataset. Although new theories are emerging to face

the challenge of model evolutions (Demaeyer and Vannitsem, 2020), several consistent recent studies show that the length of the calibration period is more critical than the strict homogeneity of datasets to forecast rare events (Lang et al., 2020; Hess, 2020). In the case of HN forecasts from EMOS (Nousu et al., 2019), even a 4-year calibration period was detrimental for the reliability of severe snowfall events compared to a longer heterogeneous reforecast. However, Taillardat and Mestre (2020) manage to successfully implement QRF in real-time forecasting products of hourly precipitation using a calibration limited

to 2-year operational forecasts, because they adapt the distribution tail with a parametric method. Producing reforecasts more homogeneous with operational forecasts is still one of the most promising solution to improve the forecast probabilities of severe events but the evolutive skill of NWP systems is strongly linked to the available data to assimilate and will never be completely removed. Therefore, the robustness of post-processing algorithms for their transfer to operational dataset or their efficiency when calibrated with shorter datasets will always remain the most important criteria compared to their theoretical

added values with perfect and long datasets. This is therefore a major point to consider to transpose the advances of this paper towards operational automatic HN forecasts.



*Code and data availability.* The R code used for the application of the EMOS approach is based on different scripts originally developed by Michael Scheuerer (Cooperative Institute for Research in Environmental Sciences - University of Colorado Boulder - and NOAA Earth System Research Laboratory, Physical Sciences Division, Boulder-Colorado, USA). The modified version can be provided on request, with the agreement of the original author. The Crocus snowpack model is developed inside the open-source SURFEX project (http://www.umr-cnrm.fr/surfex/, last access: 22 April 2021). The most up-to-date version of the code can be downloaded from the specific branch of the git repository maintained by the Centre d'Etudes de la Neige. For reproducibility of results, the version used in this work is tagged as "s2m_reanalysis_2018" on the SURFEX git repository (git.umrcnrm.fr/git/Surfex_Git2.git, last access: 22 April 2021). The full procedure and documentation to access this git repository can be found at https://opensource.cnrm-game-meteo.fr/projects/snowtools_git/wiki (last access: 22 April 2021). The codes of PEARP and SAFRAN are not currently open source. For reproducibility of results, the PEARP version used in this study is "cy42_peace-op2.18" and the SAFRAN version is tagged as "reforecast_2018" in the private SAFRAN git repository. The raw data of HN forecasts and reforecasts of the PEARP-S2M system can be obtained on request. The HN observations used in this work are public data available at https://donneespubliques.meteofrance.fr (last access: 22 April 2021).

## Appendix A: Ensemble model output statistics for post-processing of ensemble forecasts of the daily HN

### A1    Zero-censored Censored Shifted-Gamma regression

Here, the Zero-censored Censored Shifted-Gamma regression distribution (CSGD) is used to represent the predictive distribution of daily HN forecasts, and is defined as:

$$\tilde{G}_{k,\theta,\delta}(y) = \begin{cases} G_k\big(\frac{y-\delta}{\theta}\big) & \text{for } y \geq 0, \\ 0 & \text{for } y < 0 \end{cases} \tag{A1}$$

where $k$, $\theta$ and $\delta$ are shape, scale and shift parameters, respectively, and $G_k$ is the CDF of a standard gamma distribution with unit scale and shape parameter $k$. The shape parameter $k$ and scale parameter $\theta$ are directly related to the mean $\mu$ and the standard deviation $\sigma$ of the gamma distribution through the relations $\mu = k\theta$ and $\sigma^2 = k\theta^2$. Scheuerer and Hamill (2018) propose a non-homogeneous regression based on the CSGD which combines a CSGD representing he climatology of past observations. For a given day, the parameters $\mu$, $\sigma$ and $\delta$ of the predictive CSGD are related to the climatology and to the raw forecast ensemble with the following expressions (Scheuerer and Hamill, 2018, Section 3.a.):

$$\mu = \frac{\mu_{\text{cl}}}{\alpha_1}\text{log1p}\big[\text{expm1}(\alpha_1)\big(\alpha_2 + \alpha_3\text{POP} + \alpha_4\bar{x}\big)\big], \tag{A2}$$

$$\sigma = \sigma_{\text{cl}}\big(\beta_1\sqrt{\frac{\mu}{\mu_{\text{cl}}}} + \beta_2\text{MD}\big), \tag{A3}$$

$$\delta = \delta_{\text{cl}}, \tag{A4}$$

where $\text{log1p}(u) = \log(1 + u)$, and $\text{expm1}(u) = \exp(u) - 1$. The shift parameter $\delta$ is fixed at its climatological value $\delta_{\text{cl}}$. This regression model only employs the statistical properties of HN ensemble forecasts, summarized by its ensemble mean $\bar{x}$, the





probability of having a positive value POP, and the ensemble mean difference MD (a metric of ensemble spread), as defined by the following equations:

$$\bar{x} = \frac{1}{M} \sum_{m=1}^{M} x_m, \tag{A5}$$

$$\text{POP} = \frac{1}{M} \mathcal{I}_{x_m > 0}, \tag{A6}$$

$$\text{MD} = \frac{1}{M^2} \sum_{m=1}^{M} \sum_{m'=1}^{M} |x_m - x_{m'}|, \tag{A7}$$

with $x_m$ the raw HN forecast of each member $m$ among the $M$ members, and $\mathcal{I}_{x_m > 0} = 1$ if $x_m > 0$, and 0 otherwise.

### A2 Parameter estimation

For each station, the 6 parameters $\{\alpha_1, \alpha_2, \alpha_3, \alpha_4, \beta_1, \beta_2\}$ in Eqs A2-A4 are estimated by optimizing the CRPS prediction skill on the training dataset. As CRPS can be directly expressed in the case of a CSG distribution (when $F_i = \tilde{G}_{k,\theta,\delta}$), this score can be easily minimized for this EMOS model. Complete expressions of the CRPS and details about model fitting are given in Scheuerer and Hamill (2015).

### A3 Predictive distribution

Using Eqs A2-A4, parameter estimates $\{\hat{\alpha}_1, \hat{\alpha}_2, \hat{\alpha}_3, \hat{\alpha}_4, \hat{\beta}_1, \hat{\beta}_2\}$ obtained on the training dataset, and summary statistics $\bar{x}$, POP and MD of a new ensemble forecasts, we directly obtain $\hat{\mu}$, $\hat{\sigma}$ and $\hat{\delta}$. This fully specifies the predictive distribution $\tilde{G}_{\hat{k},\hat{\theta},\hat{\delta}}(y)$ of this new ensemble forecasts, with $\hat{k} = \hat{\mu}^2/\hat{\sigma}^2$ and $\hat{\theta} = \hat{\sigma}^2/\hat{\mu}$.

*Author contributions.* ML developed and ran the SURFEX-Crocus snowpack simulations forced by PEARP-SAFRAN outputs. GE set up the statistical framework, with scientific contributions of MT and MZ. GE produced the figures. GE and ML wrote the publication, with contributions of all the authors.

*Competing interests.* The authors declare that they have no conflict of interest.

*Acknowledgements.* This work has been undertaken inside the project PROSNOW which has received funding from the European Union's Horizon 2020 research and innovation program under grant agreement No 730203. The authors would like to thank Bruno Joly who developed and ran the PEARP reforecast, Matthieu Vernay who developed and ran the SAFRAN downscaling of the PEARP reforecast and real-time forecasts and Michael Scheuerer for providing the initial code of the EMOS-CSGD. CNRM/CEN and INRAE are part of LabEX OSUG@2020 (ANR10 LABX56).





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
