# Peer review of "Calibrated ensemble forecasts of the height of new snow using quantile regression forests and Ensemble Model Output Statistics"

_Nonlinear Processes in Geophysics, 2021_

## Editor Comment (EC1)

Chers Collègues,

Three referees have now submitted their reviews of your paper, and the open discussion has been closed. The three referees recommend acceptance of the paper subject to minor revisions, and make specific suggestions in that respect.

You have been requested to respond to the referees' comments by 25 August next. I encourage you, as I presume you have already started to do, to prepare a new version of the paper, taking into account the comments and suggestions of the three referees. Implementing some of these suggestions will require some work from you (for instance, referee 3's recent suggestion to implement EMOS with rainfall and temperature based predictors). But I think these suggestions are appropriate.

You may submit your revised version either with your response to the referees' comments, or at a later stage. In any case, please respond precisely at some stage to all of the referees' comments and suggestions. Should you disagree with a particular comment, or decide not to follow a particular suggestion, please state precisely your reasons for that.

I have as Editor suggestions for a few minor changes

L. 2, *Météo-France* (with accents)

L. 111, *CDF* not defined at this stage. Give the expansion there.

L. 191, *An important observed HN* …. For what I know, the word *important* cannot in English mean *numerically large.* Change to *A large observed* … (similarly, on l. 10, I suggest you change to *The gain of performance is large* … or *is significant* …)

---

## Editor Comment (EC2)

Comment 3.3 of Reviewer #3 mentions sharpness of the probability distributions produced by the various algorithms considered in the paper. The ROC curves shown on Fig. 7 are actually diagnostics of sharpness. They show the extent to which the obtained pdf's are close to the conditions of systematic 'hits' and systematic absence of 'false alarms' (top-left corner of the panels). Those conditions are those of perfect forecasts, *i.e.* perfect sharpness. The figure shows that, by that measure, QRF is at least as good as the other algorithms.

---

## Author Comment (AC2)

**Reviewer #2**

*RC#2.1. This is a useful paper which shows very clear benefits from calibration of ensemble forecasts of snow depth. Two calibration methods are compared and the quantile regression method shows clear advantages over the more standard EMOS approach (although it should perhaps be noted that there are many ways of implementing EMOS and other approaches could perform better than the one used here). It is good also that the authors have included the section 7.3 on the limitations for operational use. This is an important factor, that many calibration methods are mathematically skilful but not practical to apply for real-world forecasting, often due to lack of suitable training data, so it is good to discuss this openly in the paper. I would recommend publication with only minor amendments.*

*I have suggested to Editor that the title is not understandable to a wide audience (see comment 1 below). I have also suggested that not all figures are of appropriate quality (see note 6 below.)*

We thank the reviewer for these positive comments and suggestions. We answer below.

*RC#2.2. For me the term "height of new snow" is confusing. I suspect this is simply a slight mistranslation from the authors' native French, but causes confusion to a native English speaker. The normal English term would be "depth of snow", whereas "height" would be used more for the altitude (height up the mountains for example) where snow would occur. (A Google search for "height of new snow" returns many references to depth of snow.) I would recommend changing the word height to depth throughout the paper, including the title, and hence also HN to DN. However, I do note from the references that the authors have published a previous paper on the topic using the same term "height of new snow", so I would understand if they want to keep it for consistency. In this case, it would be worth defining what they mean clearly in the Introduction to avoid confusion.*

The use of the term "height of new snow" was asked by a reviewer of a previous publication (Vernay et al., 2015) because this is the official name for this variable in the International Classification for Seasonal Snow on the Ground of the International Association of Cryospheric Sciences (IACS) (Fierz et al., 2009). Therefore we now apply this recommendation in all our publications for homogeneity. We will add the reference at the first occurence of the term in the introduction.

*RC#2.3. Line 34: Delete the word "from": "…This prevents an appropriate correction…"*

Thank you, this will be corrected in the revised manuscript.

*RC#2.4. Table 1: Abbreviation IQR is not defined – assume it is Inter-quartile Range – but should be defined.*

This will be added.

*RC#2.5. It is interesting that all the predictors used are univariate ensemble summary statistics which means that correlations between variables present in the ensemble members are lost. This might be worth some mention – it is very encouraging that the*

*methods are successful, but it might be expected that some higher skill might be achievable if correlations between for example precipitation and Near-surface temperature were retained. Might be worth comment.*

Thank you for this interesting comment. What we understand is that the correlations between the different variables for the same ensemble could be exploited to improve our prediction. Computing the cross-correlations between some variables could indeed be an option. Note that a closely related perspective could consist in computing additional predictors based on several variables present in the forecasts (as done in, e.g., Zamo et al., 2014; Whan and Schmeits , 2018). The choice of the most relevant combinations in our case remains an open question though. If the reviewer has a specific reference in mind, we will be pleased to add it to the discussion.

*RC#2.6. Line 158: The notation used for the intervals looks odd, with opening square brackets at both ends. In figure 6 the closing at the end of the interval uses a closing round bracket, which looks better.*

Thank you for this suggestion, this will be modified.

*RC#2.7. Figures 3 and 4: I found the colours difficult to interpret when they are overlaid. It does become easier in combination with the text description, but I would suggest some alternative which clarifies the ranges for each colour. Perhaps you could mark the upper and lower bounds (10th and 90th) of each shading with overlaid lines in strong colours. (Also, this would be much worse for someone who is colour blind and cannot distinguish red and green – a different set of colours would be better but if you add lines as suggested then they could also use different line patterns.)*

Thank you for this suggestion. We will change the colors to suit colorblind safe colors and add lines of different types (e.g. plain, dotted, dashed) as suggested. It is also asked by the other reviewer.

*RC#2.8. Line 200: "QRF leads to an improvement …" – technically the plots show that EMOS and RAW are degraded relative to QRF.*

Ok, we will change this comment to "*For most of the stations, EMOS shows a degradation of the performances between 20% and 30%, up to 40% compared to QRF*".

*RC#2.9. Line 210: The term ROC has not been defined "Relative Operating Characteristic" (or alternative versions of the name). Also, you do describe ROC here briefly in lines 210-212, but why did you not define it in section 5 where all the other evaluation scores are defined?*

Thank you for this comment. As suggested, for the sake of consistency, we will move this paragraph to the evaluation section. We will also add the definition of the term ROC "Relative Operating Characteristic".

*RC#2.10. Line 216: You are describing the blue curve here, not the red one.*

Thank you for noticing this mistake, this will be corrected.

*RC#2.11. Line 316: "he" should be "the"*

Thank you very much for noticing this typo. "he climatology" will be changed to "the climatology".

*References*

Fierz, C., R. Armstrong, Y. Durand, P. Etchevers, E. Greene, D. Mcclung, K. Nishimura, P. Satyawali, and S. Sokratov. 2009. "The International Classification for Seasonal Snow on the Ground (UNESCO, IHP (International Hydrological Programme)–VII, Technical Documents in Hydrology, No 83; IACS (International Association of Cryospheric Sciences) Contribution No 1)."

Vernay, M., M.Lafaysse, L. Mérindol, G. Giraud, and S. Morin. 2015. "Ensemble Forecasting of Snowpack Conditions and Avalanche Hazard." *Cold Regions Science and Technology* 120 (December): 251–62. https://doi.org/10.1016/j.coldregions.2015.04.010.

Whan, K., and M. Schmeits. 2018. "Comparing Area Probability Forecasts of (Extreme) Local Precipitation Using Parametric and Machine Learning Statistical Postprocessing Methods." *Monthly Weather Review* 146 (11): 3651–73. https://doi.org/10.1175/MWR-D-17-0290.1.

Zamo, M., O. Mestre, P. Arbogast, and O. Pannekoucke. 2014. "A Benchmark of Statistical Regression Methods for Short-Term Forecasting of Photovoltaic Electricity Production, Part I: Deterministic Forecast of Hourly Production." *Solar Energy* 105 (July): 792–803. https://doi.org/10.1016/j.solener.2013.12.006.

---

## Author Comment (AC3)

**Reviewer #3**

*RC#3.1. The manuscript "Calibrated ensemble forecasts of the height of new snow using quantile regression forests and Ensemble Model Output Statistics" demonstrates the advantages of quantile regression forests (QRF) for postprocessing of the height of new snow. The authors perform an in-depth comparison with ensemble model output statistics (EMOS) in terms of forecast skill and discuss important aspects of QRF with regard to operational implementations. As the paper is well written and its usefulness is clearly motivated, I only have very few comments.*

We thank the reviewer for this positive feedback, for these constructive comments and for raising the numerous technical corrections that are required.

*RC#3.2. The results suggest that rainfall related predictors improve forecast skill for QRF, probably also because of poor prediction of the snow/rain limit. As the implemented version of EMOS does not use any rainfall (and temperature) related predictors, QRF outperforms EMOS considerably in situations of rainfall, but zero snow, forecasts. For a fairer comparison, I would suggest constructing a second EMOS reference that uses also rainfall (and temperature) based predictors, possibly including interactions. This would help to evaluate, if the increase in skill by QRF really comes from advantageous properties of QRF compared to EMOS or just by the fact that QRF considers more predictors.*

Thank you for this comment. We agree that the EMOS version evaluated in Nousu et al. (2019) on HS data and used as a benchmark in our paper is not able to provide correct post-processed forecasts when the snow/rain limit is not predicted accurately, since it does exploit the information provided by other predictors. In our knowledge, there is just one extended version of EMOS-CSGD using more than one ensemble of forecasts (i.e. two or more variables are used as predictors), as proposed by Scheuerer and Hamill (2015). Scheuerer and Hamill (2015) propose to use the precipitable water in addition to the precipitation forecast. This extension could be adapted to our case to add more terms in Eq. A2 and A3 of the current paper. Precisely, these linear combinations can integrate more predictors such as rainfall and temperature forecasts in order to obtain a mean and standard deviation of CSGD that integrate the information about these predictors. We implemented and tested this version, the results in terms of CRPS being shown in Figure 2 below. This version does not lead to an improvement of the CRPS

scores.

[Figure]

Figure 2: Boxplots of CRPS (left plots) and relative CPRS with QRF as a reference (right plots) with the different methods (Raw, EMOS, EMOS-Ext and QRF) for all locations, for a 1-day lead time. EMOS-Ext corresponds to the extension of the EMOS-CSGD method proposed by Scheuerer and Hamill (2015) where forecasts of rainfall rate and temperature are added to the predictors in addition to the statistics derived from ensembles of HN forecasts.

In this study, the EMOS version proposed by Nousu et al. (2019) was preferred because it has been extensively tested in our region, and we know well its limitations / advantages. It is used operationally in Météo-France. The main purpose of the study presented in the submitted manuscript is to prove that QRF methods could be a valid alternative against this method. However, we will acknowledge in the revised manuscript (in Section 7.1) that different extensions of this EMOS method could also be developed, implemented and tested in order to include more predictors (in particular the boosting extension of EMOS as discussed in the comment RC#1.8).

*RC#3.3. An evaluation of forecast sharpness is missing. I appreciate that the authors show not only CRPS values, but also rank histograms as a measure of calibration. However, as forecast quality is determined by both calibration and sharpness, I would suggest adding a figure that compares the sharpness of the raw ensemble, EMOS, and QRF.*

Indeed, the current manuscript does not present any criteria that assess the sharpness of the raw and post-processed forecasts. Following Gneiting et al. (2007), we will add a Table

presenting the mean width of the predictive intervals (50% and 90% nominal coverages) and boxplots showing the dispersion of these widths for all approaches.

*RC#3.3. Figures 3 and 4 are difficult to read because of overlapping intervals. I would suggest modifying the figures such that the limits of the prediction intervals of all forecasts are visible.*

Thank you for this comment that was also raised by the two other reviewers (see comment RC#2.7). We will add lines of different types and colors to highlight the different intervals.

*RC#3.4. p3/l59: useless question mark*

This question mark was actually a missing reference, this problem will be fixed.

*RC#3.5. p3/l66: "…statistics of other…"*

Thank you, this will be corrected.

*RC#3.6. p4/l72: small leading 'e' in ensemble*

Ok.

*RC#3.7. p4/l75: same leading 'r' in regression*

Ok.

*RC#3.8. p4/l76: maybe "at zero" instead of "in zero"?*

Thank you, "at zero" is better indeed.

*RC#3.9. p4/l77: small letters " zero-censored censored shifted-gamma distribution"*

Ok.

*RC#3.10. p5/l94 to 99: Don't we minimize the within group variance when maximizing homogeneity.*

Thank you for this comment. We agree that this paragraph can be confusing since we indicate that we want to maximize the homogeneity (which is correct) and then we define the homogeneity as a sum of variances that we want to be the smallest possible. We will rephrase this part in the revised manuscript.

*RC#3.11. p6/l121: "…22 seasons, one…"? as "the" sounds odd to me here*

We agree, this will be modified.

*RC#3.12. p6/l126-127: only if you apply score decomposition. Or, is the word "simultaneously" missing here.*

Yes, we meant simultaneously. This will be added.

*RC#3.13. p7/l132: "…variable and equals…"*

Thanks, "is" will be removed.

*RC#3.14. p7/l136: What do you mean by "technical"?*

We mean difficult, not straightforward. "Difficult" might be more adequate here.

*RC#3.15. p8/l165: CRPS*

Thanks, this will be corrected.

*RC#3.16. p8/l172: "…variable, i.e. node…"*

Ok.

*RC#3.17. Figure 2: I would make clearer that you are analysing the results for 24h accumulations of new snow depths here. Maybe, 1-24h, 25-48h, etc. in the subpanels' titles.*

Ok, this will be done.

*RC#3.18. p10/l190: From Figure 4 I guess that the station ID of Saint-Paul-sur-Ubaye should be 4193400.*

Thank you for noticing this error, this will be corrected.

*RC#3.19. p13/l199: Please do not mention significance without having performed any statistical hypothesis test. Either write, e.g., "considerable" or apply a statistical hypothesis test.*

We agree, "significance" will be replaced by "considerable".

*RC#3.20. p14/l203: "…leading to U-shape…."*

Thanks, this will be corrected.

*RC#3.21. p16/l216: Probably blue instead of read*

Thanks, that was a mistake.

*RC#3.22. caption of Figure 8: 2nd word: CRPS*

Ok.

*RC#3.23. caption of figure 8: Definitions in the caption and the titles of the subpanels for subpanels (b) and (c) are interchanged.*

Thank you for noticing this error, it will be corrected.

*RC#3.24. p18/l261 "adiabatic" instead of "adiabiatic"*

Ok.

*RC#3.25. p18/l265: Do you mean "statistical post-processing" instead of "statistical processing"*

Yes, that what we mean, this will be modified.

*RC#3.26. p19/l266: What is "iso-\theta_{w}^{'}"? The 1-degree Celsius isothermal level in terms of pseudo-adiabatic wet-bulb temperature?*

The pseudo-adiabatic wet-bulb temperature is the temperature that would have an air particle after an adiabatic cooling until water saturation level and then an adiabatic compression until the 1000 hPa level. The definition is available in *Compendium of Meteorology - for use by class I and II Meteorological Personnel: Volume I, part 2 - Physical meteorology, WMO, 1973, page 122, available at* https://library.wmo.int/index.php?lvl=notice_display&id=7078#.YPVtZKaxVhE The reference will be added in the revised manuscript.

*RC#3.27. p19/l273: "…with statistical post-processing…" w/o the "a"?*

The "a" will be removed.

*References*

Gneiting, T., F. Balabdaoui, and A. E. Raftery. 2007. "Probabilistic Forecasts, Calibration and Sharpness." *Journal of the Royal Statistical Society: Series B (Statistical Methodology)* 69 (2): 243–68. https://doi.org/10.1111/j.1467-9868.2007.00587.x.

Nousu, Jari-Pekka, Matthieu Lafaysse, Matthieu Vernay, Joseph Bellier, Guillaume Evin, et Bruno Joly. « Statistical Post-Processing of Ensemble Forecasts of the Height of New Snow ». *Nonlinear Processes in Geophysics* 26, nᵒ 3 (26 septembre 2019): 339‑57. https://doi.org/10.5194/npg-26-339-2019.

Scheuerer, Michael, et Thomas M. Hamill. « Statistical Postprocessing of Ensemble Precipitation Forecasts by Fitting Censored, Shifted Gamma Distributions ». *Monthly Weather Review* 143, nᵒ 11 (1 septembre 2015): 4578‑96. https://doi.org/10.1175/MWR-D-15-0061.1.

WMO. *Compendium of meteorology - for use by class I and II Meteorological Personnel: Volume I, part 1 - Dynamic meteorology*. Publications of blue series, volume 1 (1955-1984) - Education and Training Programme (2004), 1973.

---

## Author Response (AR1)

Response to reviewer's comments on

**npg-2021-18**

"Return levels and conditional distributions according to snow and meteorological factors for extreme avalanche cycles."

29 July, 2021

We thank the Editor for considering our paper for publication in Nonlinear Processes in Geophysics, as well as the three reviewers for their constructive comments and suggestions. These comments are reported in *blue* and in italic font. Most of the suggestions have been taken into account and the manuscript has been revised accordingly.

**EC**

*EC#1: Three referees have now submitted their reviews of your paper, and the open discussion has been closed. The three referees recommend acceptance of the paper subject to minor revisions, and make specific suggestions in that respect.*

*You have been requested to respond to the referees' comments by 25 August next. I encourage you, as I presume you have already started to do, to prepare a new version of the paper, taking into account the comments and suggestions of the three referees. Implementing some of these suggestions will require some work from you (for instance referee 3's recent suggestion to implement EMOS with rainfall and temperature based predictors). But I think these suggestions are appropriate.*

We tried, as much as possible, to include these suggestions. We also implemented a new version of EMOS which includes rainfall and temperature predictors, but the results were not convincing and probably not mature enough to be presented in the paper.

I have as Editor suggestions for a few minor changes

*EC#2: L. 2, Météo-France (with accents)*

Thank you, this has been done.

*EC#3: L. 111, CDF not defined at this stage. Give the expansion there.*

We define the acronym at l. 114 of the revised manuscript.

*EC#4: L. 191, An important observed HN … For what I know, the word important cannot in English mean numerically large. Change to A large observed .. (similarly, on l. 10, I suggest you change to The gain of performance is large .. or is significant …).*

Thank you for this comment. The word important has been replaced by large where it was relevant.

*EC#5: Comment 3.3 of Reviewer #3 mentions sharpness of the probability distributions produced by the various algorithms considered in the paper. The ROC curves shown on Fig. 7 are actually diagnostics of sharpness. They show the extent to which the obtained pdf's are close to the conditions of systematic 'hits' and systematic absence of 'false alarms' (top-left corner of the panels). Those conditions are those of perfect forecasts, i.e. perfect sharpness. The figure shows that, by that measure, QRF is at least as good as the other algorithms.*

Thank you for this comment about sharpness. In response to the comment 3.3 of Reviewer #3, a table has been added to the manuscript (Table 2, p. 13), which provides an additional diagnostic. Overall, it indicates that the width of the predictive intervals provided by EMOS and QRF are similar.

**Reviewer #1**

*RC#1.1. The authors propose a quantile regression forest (QRF)-based postprocessing method for the height of new snow (HN). The results are compared to a recently proposed ensemble model output statistics (EMOS) approach for postprocessing HN forecasts. QRF shows clear improvements over the EMOS model, in particular the inclusion of additional predictor variables seems to be benefitial.*

*Overall, I found the paper to be interesting, well-written and easy to follow throughout. I only have some minor and technical comments that are outlined below.*

We thank the reviewer for this positive feedback and these useful comments.

*RC#1.2. I found the description of the forecast distribution in Section 4.3 a bit confusing. Perhaps it would help to specifically clarify here that the resulting forecast is a set of quantiles derived from the observations from the final nodes, and not the empirical distribution in equation (1).*

Thank you for this relevant comment. It is true that the empirical distribution given in Eq. 1 of the current manuscript does not correspond to the implementation with the function `quantregForest`. Indeed, it is correct to indicate that the resulting forecast is a set of quantiles derived from the observations from the final nodes, and not the empirical distribution in equation (1). L. 114-120 of the revised manuscript clarify this point:

"*For QRF, the theoretical predictive distribution given a new set of predictors x is the conditional CDF introduced by Meinshausen (2006):*

$$F(y|x) \;=\; \sum_{i=1}^{n} w_i(x) \mathbf{1}(Y_i \le y) \qquad\qquad Eq.\ (1)$$

*where the weights wi(x) are deduced from the presence of Yi in a final leaf of each tree when one follows the path determined by x. In practise, the resulting forecast is a set of quantiles from $F(y|x)$ using the function `predict.quantregForest` of the package `quantregForest` in R. Different quantiles are thus computed for synthetic graphical representations or scores computations.*"

*RC#1.3. line 145 ff: Perhaps a few more details should be provided on why the CRPS is computed in this form. The motivation to create more quantile forecasts than raw ensemble members is not entirely clear to me. In the end, you compute an ensemble of quantiles that has many more members than the raw ensemble. While this makes it easier to account for the necessarily finite size of the sample from the forecast distribution and makes the comparison to EMOS (with a continuous forecast distribution) more "fair", doesn't this represent an "unfair" advantage when comparing to the raw ensemble?*

We agree that this part is complicated and maybe not sufficiently discussed. When the true forecast CDF is not fully known and represented as an ensemble of values, the CRPS is estimated with some error. Thus, using the CRPS to compare parametric probabilistic forecasts with ensemble forecasts may be misleading due to the unknown error of the estimated CRPS for the ensemble. Here, creating more quantile forecasts aim at reducing the error of the CRPS estimation, and provides a more fair comparison with the EMOS and the CRPS of the raw distribution. The raw ensemble is composed of a limited number of members (it is not a predictive distribution) and the best (the most fair) estimate is obtained using Eq. (3). An artificial augmentation of the raw ensemble is not relevant in that case because the raw ensemble does not contain more information than what is contained in the different members. This is the difference with QRF forecasts which provide a much larger set of different quantiles (usually hundreds of them) but there is no easy way to know precisely the corresponding order. The method proposed in Zamo and Naveau (2018) consists in computing a reasonable number of quantiles with a regular order and to apply an interpolation between these quantiles in order to be as close as possible to the true CRPS value that would be obtained if all predicted quantiles were known. The different CRPS computations can thus be explained by the information contained in the corresponding predicted forecasts.

More details have been provided at l. 148-153 of the revised manuscript to explain this.

*RC#1.4. line 169 ff: Compared to the description of the QRF model, the description of feature importance is rather short and probably difficult to understand for readers unfamiliar with QRF. Perhaps a few more details and fomulas here might help make this more clear.*

The notion of feature importance has been developed at l. 185-191 of the revised version.

*RC#1.5. Section 6, first paragraph: Are the CRPS values computed for the test set, the training set, or another validation period?*

Thank you for this comment. These sensitivity tests have been carried out on the test set. It is now specified (l. 181).

*RC#1.6. Figure 2: Given that the show forecasts seem to be of particular importance, wouldn't include more summary statistics from that variable further improve results?*

This is a point that could be tested. But first, it must be noted that snow rate forecasts are very correlated with HN forecasts simulated by Crocus. Very likely, including more statistics for the snow rate just adds more redundancy in the predictors. The first selection of the predictors for the snow and rain rates was based on our knowledge about the most important variables for the prediction of HN.

To verify this point, an additional experiment was performed. In addition to the approach tested in the manuscript, we test the QRF approach with 8 predictors based on snow and rate forecasts. For both snow and rain rates, we add the mean, the standard deviation, the probability to be non-zero, and the interquartile range of the forecast ensemble. Figure 1 below illustrates the corresponding results in terms of CRPS and CRPSS, similarly to Figure 5 of the manuscript. Clearly, adding more predictors does not

change the overall performance of the QRF approach, with the same average performance, and a very small variability according to the stations (see the small range of the CRPSS for the case "QRF+Pred").

[Figure]

Figure 1. Boxplots of CRPS (left plots) and relative CPRS with QRF with the predictors chosen in the manuscript as a reference (right plots) with the different methods (Raw ensemble, EMOS and QRF+Pred corresponding to the QRF with more snow and rain rates predictors) for all locations, for a 1-day lead time.

*RC#1.7. Figure 3: I find the confidence intervals difficult to distinguish due to the overlap and would suggest to split up the plot into three panels for all of the 3 models.*

The first version of this plot was considering separate plots but it appears that the comparison becomes difficult. Following a suggestion made by the other reviewer (see comment RC#2.7), we now overlay colored lines instead, in order to highlight the lower and upper bounds of each approach (see new Figures 3 and 4).

*RC#1.8. Overall, the paper in particular demonstrates that the inclusion of additional predictor variables improves performance. This is very much in line with the previous work on QRF and also several other machine learning-based postprocessing methods (for example proposed in Messner et al (2017, https://doi.org/10.1175/MWR-D-16-0088.1), Rasp and Lerch (2018, https://doi.org/10.1175/MWR-D-18-0187.1), Bremnes (2020, https://doi.org/10.1175/MWR-D-19-0227.1), and others). Since EMOS only uses forecasts of the variable of interest as predictor, it would have been more "fair" to compare the the boosting extension of EMOS proposed in the paper by Messner et al. While this is beyond the scope of the paper in the current form, I'd suggest to include this aspect in the discussion as an avenue for future work.*

Thank you for this discussion. Indeed, a boosting extension of EMOS could in principle be applied in our context. Recently, Schulz and Lerch (2021) compares a gradient-boosting extension of EMOS (EMOS-GB) to many machine learning methods

for postprocessing ensemble forecasts of wind gusts, using a truncated logistic distribution. The performances of EMOS-GB were honorable, but, as you indicate, other machine learning-based postprocessing methods seem promising. The Distributional Regression Network of Rasp and Lerch (2018) and the Bernstein Quantile Network of Bremnes (2020) often outperform all the other methods in Schulz and Lerch (2021), including QRF. This has been added to the discussion at l.254-271 (see also our response to the comment RC#3.2).

*RC#1.9. line 59: Missing reference?*

Thank you. There was indeed an issue with a reference. It has been removed.

*RC#1.10. Section 4.2: I'd suggest to consider moving this Section to the beginning of Section 6. In the current form, the meaning of the hyperparmeters mtry and nodesize is not yet explained and will be difficult to understand for readers not familiar with QRF.*

Thank you for this relevant comment. As Section 4.2 belongs to methodological aspects and are not results, we keep this paragraph here, but the revised manuscript nowl provides the definitions of mtry and nodesize in Sec. 4.2, as we agree that it cannot be understood in the current form.

*RC#1.11. Code and data availability: The limitations on availability of the EMOS and NWP model code are clearly explained, but it is unclear to me whether (or where) the QRF code is available.*

This has been added (l.329-331). We essentially use available R packages for the QRF method.

*References*

Bremnes, J. B.. 2020. "Ensemble Postprocessing Using Quantile Function Regression Based on Neural Networks and Bernstein Polynomials." *Monthly Weather Review* 148 (1): 403–14. https://doi.org/10.1175/MWR-D-19-0227.1.

Rasp, S., and S. Lerch. 2018. "Neural Networks for Postprocessing Ensemble Weather Forecasts." *Monthly Weather Review* 146 (11): 3885–3900. https://doi.org/10.1175/MWR-D-18-0187.1.

Schulz, B., and S. Lerch. 2021. "Machine Learning Methods for Postprocessing Ensemble Forecasts of Wind Gusts: A Systematic Comparison." *ArXiv:2106.09512 [Physics, Stat]*, June. http://arxiv.org/abs/2106.09512.

**Reviewer #2**

*RC#2.1. This is a useful paper which shows very clear benefits from calibration of ensemble forecasts of snow depth. Two calibration methods are compared and the quantile regression method shows clear advantages over the more standard EMOS approach (although it should perhaps be noted that there are many ways of implementing EMOS and other approaches could perform better than the one used here). It is good also that the authors have included the section 7.3 on the limitations for operational use. This is an important factor, that many calibration methods are mathematically skilful but not practical to apply for real-world forecasting, often due to lack of suitable training data, so it is good to discuss this openly in the paper. I would recommend publication with only minor amendments.*

*I have suggested to Editor that the title is not understandable to a wide audience (see comment 1 below). I have also suggested that not all figures are of appropriate quality (see note 6 below.)*

We thank the reviewer for these positive comments and suggestions. We answer below.

*RC#2.2. For me the term "height of new snow" is confusing. I suspect this is simply a slight mistranslation from the authors' native French, but causes confusion to a native English speaker. The normal English term would be "depth of snow", whereas "height" would be used more for the altitude (height up the mountains for example) where snow would occur. (A Google search for "height of new snow" returns many references to depth of snow.) I would recommend changing the word height to depth throughout the paper, including the title, and hence also HN to DN. However, I do note from the references that the authors have published a previous paper on the topic using the same term "height of new snow", so I would understand if they want to keep it for consistency. In this case, it would be worth defining what they mean clearly in the Introduction to avoid confusion.*

The use of the term "height of new snow" was asked by a reviewer of a previous publication (Vernay et al., 2015) because this is the official name for this variable in the International Classification for Seasonal Snow on the Ground of  the International Association of Cryospheric Sciences (IACS) (Fierz et al., 2009). Therefore we now apply this recommendation in all our publications for homogeneity. We have added the reference at the first occurence of the term in the introduction (l. 16).

*RC#2.3. Line 34: Delete the word "from": "…This prevents an appropriate correction…"*

Thank you, this has been corrected.

*RC#2.4. Table 1: Abbreviation IQR is not defined – assume it is Inter-quartile Range – but should be defined.*

This has been added.

*RC#2.5. It is interesting that all the predictors used are univariate ensemble summary statistics which means that correlations between variables present in the ensemble members are lost. This might be worth some mention – it is very encouraging that the*

*methods are successful, but it might be expected that some higher skill might be achievable if correlations between for example precipitation and Near-surface temperature were retained. Might be worth comment.*

Thank you for this interesting comment. What we understand is that the correlations between the different variables for the same ensemble could be exploited to improve our prediction. Computing the cross-correlations between some variables could indeed be an option. Note that a closely related perspective could consist in computing additional predictors based on several variables present in the forecasts (as done in, e.g., Zamo et al., 2014; Whan and Schmeits , 2018). The choice of the most relevant combinations in our case remains an open question though.

*RC#2.6. Line 158: The notation used for the intervals looks odd, with opening square brackets at both ends. In figure 6 the closing at the end of the interval uses a closing round bracket, which looks better.*

Thank you for this suggestion, this has been modified (see l. 171).

*RC#2.7. Figures 3 and 4: I found the colours difficult to interpret when they are overlaid. It does become easier in combination with the text description, but I would suggest some alternative which clarifies the ranges for each colour. Perhaps you could mark the upper and lower bounds (10th and 90th) of each shading with overlaid lines in strong colours. (Also, this would be much worse for someone who is colour blind and cannot distinguish red and green – a different set of colours would be better but if you add lines as suggested then they could also use different line patterns.)*

Thank you for this suggestion. The colors of these figures are now colorblind safe colors and include lines of different types (plain, dotted, dashed) as suggested.

*RC#2.8. Line 200: "QRF leads to an improvement …" – technically the plots show that EMOS and RAW are degraded relative to QRF.*

This comment has been modified: "*For most of the stations, EMOS shows a degradation of the performances between 20% and 30%, up to 40% compared to QRF*" (l. 217).

*RC#2.9. Line 210: The term ROC has not been defined "Relative Operating Characteristic" (or alternative versions of the name). Also, you do describe ROC here briefly in lines 210-212, but why did you not define it in section 5 where all the other evaluation scores are defined?*

Thank you for this comment. As suggested, for the sake of consistency, this paragraph has been moved to the evaluation section. The definition of the term ROC "Relative Operating Characteristic" has been added (l. 174-176).

*RC#2.10. Line 216: You are describing the blue curve here, not the red one.*

Thank you for noticing this mistake, this has been corrected.

*RC#2.11. Line 316: "he" should be "the"*

Thank you very much for noticing this typo. "he climatology" has been changed to "the climatology".

*References*

Fierz, C., R. Armstrong, Y. Durand, P. Etchevers, E. Greene, D. Mcclung, K. Nishimura, P. Satyawali, and S. Sokratov. 2009. "The International Classification for Seasonal Snow on the Ground (UNESCO, IHP (International Hydrological Programme)–VII, Technical Documents in Hydrology, No 83; IACS (International Association of Cryospheric Sciences) Contribution No 1)."

Vernay, M., M.Lafaysse, L. Mérindol, G. Giraud, and S. Morin. 2015. "Ensemble Forecasting of Snowpack Conditions and Avalanche Hazard." *Cold Regions Science and Technology* 120 (December): 251–62. https://doi.org/10.1016/j.coldregions.2015.04.010.

Whan, K., and M. Schmeits. 2018. "Comparing Area Probability Forecasts of (Extreme) Local Precipitation Using Parametric and Machine Learning Statistical Postprocessing Methods." *Monthly Weather Review* 146 (11): 3651–73. https://doi.org/10.1175/MWR-D-17-0290.1.

Zamo, M., O. Mestre, P. Arbogast, and O. Pannekoucke. 2014. "A Benchmark of Statistical Regression Methods for Short-Term Forecasting of Photovoltaic Electricity Production, Part I: Deterministic Forecast of Hourly Production." *Solar Energy* 105 (July): 792–803. https://doi.org/10.1016/j.solener.2013.12.006.

**Reviewer #3**

*RC#3.1. The manuscript "Calibrated ensemble forecasts of the height of new snow using quantile regression forests and Ensemble Model Output Statistics" demonstrates the advantages of quantile regression forests (QRF) for postprocessing of the height of new snow. The authors perform an in-depth comparison with ensemble model output statistics (EMOS) in terms of forecast skill and discuss important aspects of QRF with regard to operational implementations. As the paper is well written and its usefulness is clearly motivated, I only have very few comments.*

We thank the reviewer for this positive feedback, for these constructive comments and for raising the numerous technical corrections that were required.

*RC#3.2. The results suggest that rainfall related predictors improve forecast skill for QRF, probably also because of poor prediction of the snow/rain limit. As the implemented version of EMOS does not use any rainfall (and temperature) related predictors, QRF outperforms EMOS considerably in situations of rainfall, but zero snow, forecasts. For a fairer comparison, I would suggest constructing a second EMOS reference that uses also rainfall (and temperature) based predictors, possibly including interactions. This would help to evaluate, if the increase in skill by QRF really comes from advantageous properties of QRF compared to EMOS or just by the fact that QRF considers more predictors.*

Thank you for this comment. We agree that the EMOS version evaluated in Nousu et al. (2019) on HS data and used as a benchmark in our paper is not able to provide correct post-processed forecasts when the snow/rain limit is not predicted accurately, since it does exploit the information provided by other predictors. In our knowledge, there is just one extended version of EMOS-CSGD using more than one ensemble of forecasts (i.e. two or more variables are used as predictors), as proposed by Scheuerer and Hamill (2015). Scheuerer and Hamill (2015) propose to use the precipitable water in addition to the precipitation forecast. This extension could be adapted to our case to add more terms in Eq. A2 and A3 of the current paper. Precisely, these linear combinations can integrate more predictors such as rainfall and temperature forecasts in order to obtain a mean and standard deviation of CSGD that integrate the information about these predictors. We implemented and tested this version, the results in terms of CRPS being shown in Figure 2 below. This version does not lead to an improvement of the CRPS

scores.

[Figure]

Figure 2: Boxplots of CRPS (left plots) and relative CPRS with QRF as a reference (right plots) with the different methods (Raw, EMOS, EMOS-Ext and QRF) for all locations, for a 1-day lead time. EMOS-Ext corresponds to the extension of the EMOS-CSGD method proposed by Scheuerer and Hamill (2015) where forecasts of rainfall rate and temperature are added to the predictors in addition to the statistics derived from ensembles of HN forecasts.

In this study, the EMOS version proposed by Nousu et al. (2019) was preferred because it has been extensively tested in our region, and we know well its limitations / advantages. It is used operationally in Météo-France. The main purpose of the study presented in the submitted manuscript is to prove that QRF methods could be a valid alternative against this method. However, we acknowledge in the revised manuscript (in Section 7.1) that different extensions of this EMOS method could also be developed, implemented and tested in order to include more predictors (in particular the boosting extension of EMOS as discussed in the comment RC#1.8).

*RC#3.3. An evaluation of forecast sharpness is missing. I appreciate that the authors show not only CRPS values, but also rank histograms as a measure of calibration. However, as forecast quality is determined by both calibration and sharpness, I would suggest adding a figure that compares the sharpness of the raw ensemble, EMOS, and QRF.*

Indeed, the current manuscript does not present any criteria that assess the sharpness of the raw and post-processed forecasts. Following Gneiting et al. (2007), Table 2 of the

revised manuscript presents the mean width of the predictive intervals (50% and 90% nominal coverages) and the standard deviations of these widths (see p. 13).

*RC#3.3. Figures 3 and 4 are difficult to read because of overlapping intervals. I would suggest modifying the figures such that the limits of the prediction intervals of all forecasts are visible.*

Thank you for this comment that was also raised by the two other reviewers (see comment RC#2.7). Lines of different types and colors have been added to highlight the different intervals.

*RC#3.4. p3/l59: useless question mark*

This question mark was actually a missing reference, this problem has been fixed.

*RC#3.5. p3/l66: "…statistics of other…"*

Thank you, this has been corrected.

*RC#3.6. p4/l72: small leading 'e' in ensemble*

Ok.

*RC#3.7. p4/l75: same leading 'r' in regression*

Ok.

*RC#3.8. p4/l76: maybe "at zero" instead of "in zero"?*

Thank you, "at zero" is better indeed.

*RC#3.9. p4/l77: small letters " zero-censored censored shifted-gamma distribution"*

Ok.

*RC#3.10. p5/l94 to 99: Don't we minimize the within group variance when maximizing homogeneity.*

Thank you for this comment. We agree that this paragraph can be confusing since we indicate that we want to maximize the homogeneity (which is correct) and then we define the homogeneity as a sum of variances that we want to be the smallest possible. This part has been modified in the revised manuscript (l. 97).

*RC#3.11. p6/l121: "…22 seasons, one…"? as "the" sounds odd to me here*

We agree, this has been modified.

*RC#3.12. p6/l126-127: only if you apply score decomposition. Or, is the word "simultaneously" missing here.*

Yes, we meant simultaneously. This has been added.

*RC#3.13. p7/l132: "…variable and equals…"*

Thanks, "is" has been removed.

*RC#3.14. p7/l136: What do you mean by "technical"?*

We mean difficult, not straightforward. "Difficult" might be more adequate here.

*RC#3.15. p8/l165: CRPS*

Thanks, this has been corrected.

*RC#3.16. p8/l172: "…variable, i.e. node…"*

Ok.

*RC#3.17. Figure 2: I would make clearer that you are analysing the results for 24h accumulations of new snow depths here. Maybe, 1-24h, 25-48h, etc. in the subpanels' titles.*

Ok, this has been done.

*RC#3.18. p10/l190: From Figure 4 I guess that the station ID of Saint-Paul-sur-Ubaye should be 4193400.*

Thank you for noticing this error, this has been corrected.

*RC#3.19. p13/l199: Please do not mention significance without having performed any statistical hypothesis test. Either write, e.g., "considerable" or apply a statistical hypothesis test.*

We agree, "significance" has been replaced by "considerable".

*RC#3.20. p14/l203: "…leading to U-shape…."*

Thanks, this has been corrected.

*RC#3.21. p16/l216: Probably blue instead of read*

Thanks, that was a mistake.

*RC#3.22. caption of Figure 8: 2nd word: CRPS*

Ok.

*RC#3.23. caption of figure 8: Definitions in the caption and the titles of the subpanels for subpanels (b) and (c) are interchanged.*

Thank you for noticing this error, it has been corrected.

*RC#3.24. p18/l261 "adiabatic" instead of "adiabiatic"*

Ok.

*RC#3.25. p18/l265: Do you mean "statistical post-processing" instead of "statistical processing"*

Yes, that what we mean, this has been modified.

*RC#3.26. p19/l266: What is "iso-\theta_{w}^{'}"? The 1-degree Celsius isothermal level in terms of pseudo-adiabatic wet-bulb temperature?*

The pseudo-adiabatic wet-bulb temperature is the temperature that would have an air particle after an adiabatic cooling until water saturation level and then an adiabatic compression until the 1000 hPa level. The definition is available in *Compendium of Meteorology - for use by class I and II Meteorological Personnel: Volume I, part 2 - Physical meteorology, WMO, 1973, page 122, available at* https://library.wmo.int/index.php?lvl=notice_display&id=7078#.YPVtZKaxVhE The reference has been added in the revised manuscript (l. 295).

*RC#3.27. p19/l273: "…with statistical post-processing…" w/o the "a"?*

The "a" has been removed.

*References*

Gneiting, T., F. Balabdaoui, and A. E. Raftery. 2007. "Probabilistic Forecasts, Calibration and Sharpness." *Journal of the Royal Statistical Society: Series B (Statistical Methodology)* 69 (2): 243–68. https://doi.org/10.1111/j.1467-9868.2007.00587.x.

Nousu, Jari-Pekka, Matthieu Lafaysse, Matthieu Vernay, Joseph Bellier, Guillaume Evin, et Bruno Joly. « Statistical Post-Processing of Ensemble Forecasts of the Height of New Snow ». *Nonlinear Processes in Geophysics* 26, n° 3 (26 septembre 2019): 339-57. https://doi.org/10.5194/npg-26-339-2019.

Scheuerer, Michael, et Thomas M. Hamill. « Statistical Postprocessing of Ensemble Precipitation Forecasts by Fitting Censored, Shifted Gamma Distributions ». *Monthly Weather Review* 143, n° 11 (1 septembre 2015): 4578-96. https://doi.org/10.1175/MWR-D-15-0061.1.

WMO. *Compendium of meteorology - for use by class I and II Meteorological Personnel: Volume I, part 1 - Dynamic meteorology*. Publications of blue series, volume 1 (1955-1984) - Education and Training Programme (2004), 1973.

---

## Editor Decision (ED1)

Chers Collègues,

I have received three short reviews of the revised version of your paper. They have been submitted by the same three referees of the first version, who had all recommended minor revisions of the paper. The referees are identified by the same numbers as before (Referee #2 has in the meantime let his name known, and is Ken Mylne, from the British Met Office).

All three referees recommend acceptance of the paper, with two suggestions from Referees #1 and #2, which I suggest you follow.

I have in addition as Editor a number of comments and suggestions.

1. My first comment has actually to do, at least in part, with science. You apparently use the words *resolution* and *sharpness* as if they corresponded to different properties of a probabilistic prediction system. They actually correspond to the same property, namely (as you write) the *ability to separate a priori the probability classes*, or to distinguish *a priori* between different outcomes (see Broecker, 2014). Please use only one word or, if you use both, say they refer to the same property.

I add that the ROC curve, shown on your Figure 7, is a diagnostic of that property. It shows the degree to which the system under consideration is able to distinguish *a priori* between 'hits' and 'false alarms', *i.e.* between occurrence or non-occurrence of the considered events. That is exactly sharpness. I suggest you replace the words *'good' prediction* in the caption of the figure with the words *sharp prediction system* (that will also remove the uncertainty implied by the quotation marks in *'good'*).

2. A number of acronyms are not expanded, at least not the first time they are used (*e.g.* PEARP-S2M on l. 31). Please check systematically that all acronyms are expanded on their first occurrence, and give appropriate references whenever necessary.

3. Figure 6. Was the number of intervals used for building the histograms arbitrary, or did it correspond to anything imbedded from the start in the prediction system. If yes, to what does it correspond (that is not clear to me) ?

4. L. 202, … *at the end of  the period*.

5. Table 1, l. 7. *PR0  Raw probability of HN > 0*

6. L. 216,  … → *The right panel* …

7. I find the caption of Figure 3 somewhat confusing. I suggest … *lead time (orange full lines)* …, *QRF (purple dashes) and EMOS (green points)*. And next

sentence *For each of the three prediction systems, the lower and upper curves represent the 10th and 90th percentiles respectively*.

8. Figure 2. Say more precisely what the vertical coordinate on the figure is.

9. L. 139, … *equals 0* (or *is equal to 0*)

10. L. 219 (and Table 2). I presume *CI* means centiles?

Please revise your paper along the suggestions of Referees #1 and #2, as well as along my own. Should you disagree with a particular suggestion and decide not to follow it, please state precisely your reasons for that.

I am looking forward to receiving the revised version of your paper.

**REFERENCE**

Broecker, J. (2014) *Resolution and discrimination - two sides of the same coin*. Quarterly Journal of the Royal Meteorological Society, 141 (689). pp. 1277-1282. ISSN 1477-870X doi: https://doi.org/10.1002/qj.2434

---

## Author Response (AR2)

Response to reviewer's comments on

**npg-2021-18 v2**

"Calibrated ensemble forecasts of the height of new snow using quantile regression forests and Ensemble Model Output Statistics"

19 August, 2021

We thank the Editor and the three reviewers for their positive feedback and their comments on the revised version. These comments are reported in *blue* and in italic font. The additional suggestions have been taken into account. Please see our response to the different comments below.

**EC**

*EC#1: My first comment has actually to do, at least in part, with science. You apparently use the words resolution and sharpness as if they corresponded to different properties of a probabilistic prediction system. They actually correspond to the same property, namely (as you write) the ability to separate a priori the probability classes, or to distinguish a priori between different outcomes (see Broecker, 2014). Please use only one word or, if you use both, say they refer to the same property.*

Thank you for this relevant comment. The term "resolution" has been replaced by "sharpness" throughout the revised version, except when it was used to refer to the spatial resolution of NWP models.

*EC#2: I add that the ROC curve, shown on your Figure 7, is a diagnostic of that property. It shows the degree to which the system under consideration is able to distinguish a priori between 'hits' and 'false alarms', i.e. between occurrence or non-occurrence of the considered events. That is exactly sharpness. I suggest you replace the words 'good' prediction in the caption of the figure with the words sharp prediction system (that will also remove the uncertainty implied by the quotation marks in 'good').*

Thank you, this has been done.

*EC#3: A number of acronyms are not expanded, at least not the first time they are used (e.g. PEARP-S2M on l. 31). Please check systematically that all acronyms are expanded on their first occurrence, and give appropriate references whenever necessary.*

The meaning of PEARP-S2M and the related acronyms are now defined at l. 31-34 of the revised manuscript. Different corrections have been made for other acronyms (CPRSS replaced by CRPS, CI and IC have been replaced by PI for "predictive intervals, definition of CART")".

*EC#4: Figure 6. Was the number of intervals used for building the histograms arbitrary, or did it correspond to anything imbedded from the start in the prediction system. If yes, to what does it correspond (that is not clear to me) ?*

The number of intervals was arbitrary, not too small to have a fine description of the distribution of the ranks, but not too high in order to have a sufficient number of items inside each class.

*EC#5: . L. 202, … at the end of  the period.*

Thank you, this has been corrected.

*EC#6: Table 1, l. 7. PR0 Raw probability of HN > 0*

This has been corrected.

*EC#7: L. 216, The second column … → The right panel …*

This has been corrected.

*EC#8: I find the caption of Figure 3 somewhat confusing. I suggest … lead time (orange full lines) …, QRF (purple dashes) and EMOS (green points). And next sentence For each of the three prediction systems, the lower and upper curves represent the 10th and 90th percentiles respectively.*

The captions of Figures 3 and 4 have been modified. We now refer to "orange plain lines", "purple dashed lines" and "green dotted lines".

*EC#9: Figure 2. Say more precisely what the vertical coordinate on the figure is.*

We now precise that importance is the "sum of squares of the differences between predicted and observed response variables, averaged over all trees obtained with the random permutations". We have also corrected the related sentence before, as it was not correct to indicate that it was the standard deviation of the response variable.

*EC#10: L. 139, … equals 0 (or is equal to 0)*

This has been changed to "equals 0".

*EC#11: L. 219 (and Table 2). I presume CI means centiles?*

CI was indeed misleading and we now indicate that these ranges are related to predictive intervals (PI).

**Reviewer #1**

*Thank you for your responses and revision of the paper. I only have one remaining very minor comment. In your response, you wrote that "Thank you for this comment. These sensitivity tests have been carried out on the test set. It is now specified (l. 181)." In line 181 of the revised paper, you write that ") average CRPS values for the validation datasets" Which is correct?*

Thank you for your comment. The manuscript is correct, the average CRPS values for the sensitivity tests have been computed on the validation datasets, using the leave-one season out cross-validation scheme.

**Reviewer #2**

*Term Height of new snow – I accept the authors argument that this is a standard term in cryosphere science and the addition of reference is helpful. Nevertheless, most readers are not cryosphere specialists and I still think a very short phrase such as "…height of new snow (Fierz et al., 2009) (also commonly known as depth of snow) …." would be helpful to a large proportion of readers.*

We appreciate this comment and we have added "also commonly known as depth of fresh snow" after "Fierz et al., 2009", the term "fresh" being important to avoid confusions with the long-term snow depth (height of the snow cover).